# Policy Search via Bayesian Optimization with Temporal Difference Gaussian Processes

**Armin Lederer** [1]   **Anuj Srivastava** [2]   **Marco Bagatella** [2 3]   **Andreas Krause** [2]

## Abstract

Bayesian optimization (BO) is a method commonly used for policy search in problems with low-dimensional policy parameterizations. While it is generally considered data-efficient, existing BO approaches are agnostic to the sequential structure of the optimization objective induced by policy roll-outs. Thereby, valuable information is discarded that could improve the convergence of BO. We address this inefficiency by developing and rigorously analyzing a novel approach for BO that relies on a temporal difference learning formulation for discounted infinite-horizon value functions based on Gaussian process (GP) regression. We derive learning error bounds for the proposed temporal difference GPs, such that we can exploit upper confidence bounds to analyze the cumulative regret of our BO approach. This analysis is further refined by bounding the maximal information gain for our temporal difference GP model. In a comparison with relevant baseline methods, we demonstrate the practical advantages of our method.

## 1. Introduction

Policy search is a class of reinforcement learning (RL) methods which is particularly popular in robotics (Deisenroth, 2013; Sigaud & Stulp, 2019). Policy search relies on policy parameterizations, often comparatively low-dimensional and tailored to specific tasks, leading to success in a large variety of real-world robotics problems, such as balancing tasks (Doerr et al., 2017), stroke movements (Peters & Schaal, 2008), object manipulation (Gupta et al., 2016), and locomotion (Levine & Koltun, 2013). Among the

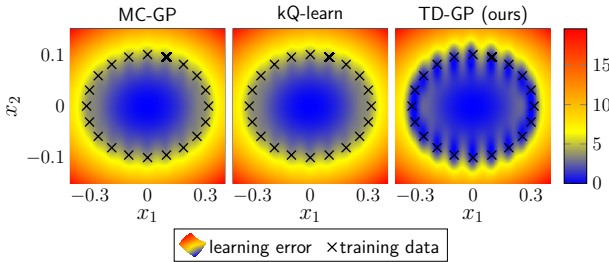

*Figure 1.* Learning accuracy for value function estimates of an inverted pendulum obtained via GP regression with Monte-Carlo target (MC-GP) (Wilson et al., 2014), GP-based value iteration as used in kernelized Q-learning (kQ-learn) (Chowdhury & Oliveira, 2023), and our proposed temporal difference GP (TD-GP). Our approach achieves low errors (blue) around data points (crosses), while existing approaches barely benefit from such short trajectories of data.

different policy search techniques, Bayesian optimization (BO) (Shahriari et al., 2016; Garnett, 2023) has gained increasing attention due its strong theoretical foundations (Srinivas et al., 2012; Chowdhury & Gopalan, 2017) as well as the straightforward consideration of safety constraints (Berkenkamp et al., 2016; 2023) and robustness requirements (Paulson et al., 2022).

BO is a class of black-box optimization algorithms that aims to find the global optimum given noisy samples of the objective function. For this purpose, it sequentially learns a model of the objective using Gaussian process (GP) regression (Rasmussen & Williams, 2006), e.g., using Monte-Carlo estimates of value functions (Wilson et al., 2014), based on which an acquisition function proxy is constructed for optimization. While BO has been demonstrated to be particularly effective in policy search problems with low-dimensional, structured parameterizations such as PID (Fiducioso et al., 2019) and LQR parameter optimization (Marco et al., 2016), existing approaches remain agnostic to the underlying sequential structure of returns induced by roll-outs. Attempts to overcome this limitation using approximate value iteration techniques can be found in related fields such as kernelized reinforcement learning (Yang et al., 2020; Chowdhury & Oliveira, 2023), where the main focus lies in learning finite horizon value functions.

[1]Department of Electrical and Computer Engineering, National University of Singapore [2]Department of Computer Science, ETH Zurich [3]Max Planck Institute for Intelligent Systems, Tuebingen. Correspondence to: Armin Lederer <armin.lederer@nus.edu.sg>.

*Proceedings of the 43rd International Conference on Machine Learning*, Seoul, South Korea. PMLR 306, 2026. Copyright 2026 by the author(s).

However, these approaches also fail to capture certain informative temporal correlations occurring in discounted infinite horizon problems as illustrated in Figure 1. Thus, existing approaches for learning value functions discard valuable information that could improve the convergence of BO in policy search problems.

**Contributions:** We address this weakness of existing BO approaches by proposing and analyzing a novel upper confidence bound algorithm that relies on a Gaussian process model of the value function learned from its temporal differences. In particular, our key contributions are the following:

- **Temporal difference GPs:** We propose a novel approach for temporal difference learning of discounted infinite-horizon value functions with GPs by exploiting the closedness of GPs under linear operators. Under weak assumptions on the transition probabilities, we extend existing probabilistic regression error bounds to the obtained value function estimate.

- **Regret bounds:** Based on the derived error bound, we extend the common upper confidence bound (UCB) approach in BO (Srinivas et al., 2012) to maximizing discounted infinite-horizon value functions. By relating the GP variances of the value function to those of temporal differences, we bound the regret of the proposed algorithm in terms of the maximum information gain.

- **Information gain analysis:** To establish regret bounds directly in terms of the number of episodes, we provide a novel analysis of the information gain given our GP model exploiting the temporal difference structure via a matrix representation. We show that our regret bounds are competitive to those of related RL approaches.

The remainder of this paper is structured as follows: We formalize our problem setting in Section 2. Our approach for policy search via BO with temporal difference GP is proposed and analyzed in Section 3. In Section 4, we discuss the connection of our method to related work. We compare our method to different baselines on benchmark problems in Section 5, before we conclude the paper in Section 6.

## 2. Problem Statement

We consider a discrete-time stochastic dynamical system

$$\boldsymbol{s}_{t+1} \sim p(\cdot|\boldsymbol{s}_t, \boldsymbol{a}_t) \qquad \boldsymbol{s}_0 \sim \rho \tag{1}$$

where $\boldsymbol{s}_t \in \mathcal{S} \subset \mathbb{R}^{d_s}$ is the state, $\boldsymbol{a}_t \in \mathcal{A} \subset \mathbb{R}^{d_a}$ is the action, $p : \mathcal{S} \times \mathcal{A} \to \Delta(\mathcal{S})$ is the transition kernel and $\rho$ is some initial state distribution. We assume that the transition kernel $p$ is unknown, but we restrict its distribution to satisfy the following properties.

**Assumption 2.1.** The transition kernel $p(\cdot|\boldsymbol{s}, \boldsymbol{a})$ is compactly supported such that $\mathcal{S}$ is a compact set. Moreover,

the process noise $\boldsymbol{w} = \boldsymbol{s}^+ - \mathbb{E}_{\boldsymbol{s}^+}[\boldsymbol{s}^+]$ is sub-Gaussian with $\sup_{\boldsymbol{v}:\|\boldsymbol{v}\|=1} \mathbb{E}_{\boldsymbol{w}}[\exp((\boldsymbol{v}^T\boldsymbol{w})^2/\lambda^2)] \leq 2$ for $\lambda \in \mathbb{R}_{>0}$.

The assumption of a compact support of $p$ ensures that trajectories generated by (1) cannot diverge indefinitely, such that learning can be realized on a bounded set. Thereby, this requirement resembles similar assumptions in literature on RL, e.g., (Sukhija et al., 2024). The restriction on noise is commonly found in literature on Bayesian optimization (Srinivas et al., 2012; Chowdhury & Gopalan, 2017) and reinforcement learning approaches (Curi et al., 2020; Kakade et al., 2020). It poses an upper bound on the decay of the tails of the distribution of $\boldsymbol{w}$. Since this assumption admits a wide range of distributions such as truncated Gaussian and uniform distributions, it is generally not considered restrictive.

We study the problem of determining a policy $\boldsymbol{\pi} : \mathcal{S} \times \Theta \to \mathcal{A}$ parameterized by $\boldsymbol{\theta} \in \Theta \subset \mathbb{R}^{d_\theta}$ with compact set $\Theta$ which maximizes the discounted average cumulative returns

$$J(\boldsymbol{\theta}) = \mathbb{E}_{\boldsymbol{s}_0}\left[V(\boldsymbol{s}_0, \boldsymbol{\theta})\right] \tag{2}$$
$$= \mathbb{E}_{\boldsymbol{s}_0}\left[\mathbb{E}_{\boldsymbol{s}_1, \boldsymbol{s}_2, \dots}\left[\sum_{t=0}^{\infty} \gamma^t r(\boldsymbol{s}_t, \boldsymbol{\pi}(\boldsymbol{s}_t, \boldsymbol{\theta}))\Big|\boldsymbol{s}_0\right]\right],$$

where $r : \mathcal{S} \times \mathcal{A} \to \mathbb{R}$ is an immediate reward, $\gamma \in (0, 1)$ is a discount factor, and the sequence of states $\boldsymbol{s}_t$ is generated by (1) with $\boldsymbol{a}_t = \boldsymbol{\pi}(\boldsymbol{s}_t, \boldsymbol{\theta})$. Since $p$ is unknown, we aim to iteratively learn the optimal policy parameter $\boldsymbol{\theta}^*$ which maximizes (2) by interacting with the dynamics (1). In particular, we consider an episodic setting in which we roll out a policy $\boldsymbol{\pi}$ with parameter $\boldsymbol{\theta}_n$ for $M_n \in \mathbb{N}$ time steps in every episode $n = 1, \dots, N$. This results in a trajectory $\tau^n = \{(\boldsymbol{s}_0^n, r_0^n), (\boldsymbol{s}_1^n, r_1^n), \dots, (\boldsymbol{s}_{M_n}^n, r_{M_n}^n)\}$ with $r_i^n = r(\boldsymbol{s}_i^n, \boldsymbol{\pi}(\boldsymbol{s}_i^n, \boldsymbol{\theta}^n))$ for each episode $n$, which we aggregate into a data set $\mathbb{D}^N = \{(\tau^n)_{n=1,\dots,N}\}$. This data set is subsequently used to learn an estimate of the returns $J$ defined in (2) based on GP regression.

The goal of this paper is the development of an algorithm for determining the parameter $\boldsymbol{\theta}_{n+1}$ for the next episode given the data $\mathbb{D}^n$ obtained so far. We measure the performance of this algorithm via the commonly used metric of cumulative regret (Sutton & Barto, 2018)

$$R_N = \sum_{n=1}^{N} J(\boldsymbol{\theta}^*) - J(\boldsymbol{\theta}_n), \tag{3}$$

where $\boldsymbol{\theta}^*$ denotes the maximizer of (2). Therefore, an intuitive requirement for a useful algorithm choosing $\boldsymbol{\theta}_n$ is $R_N/N \to 0$ for $N \to \infty$, i.e., sub-linear cumulative regret.

To formally analyze the asymptotic behavior of the cumulative regret, additional assumptions about the problem complexity are necessary. Since we employ Gaussian process regression for learning an estimate of the returns $J$, we pose this requirement in terms of the function complexity as measured through the kernel used for regression.

**Assumption 2.2.** Let $k : (\mathcal{S} \times \Theta) \times (\mathcal{S} \times \Theta) \to \mathbb{R}$ be a kernel with bounded Hessian $\mathrm{trace}\left(\nabla_{\boldsymbol{s}}\nabla_{\boldsymbol{s'}}^T k([\boldsymbol{s};\boldsymbol{\theta}],[\boldsymbol{s'};\boldsymbol{\theta}])\big|_{\boldsymbol{s'}=\boldsymbol{s}}\right) \leq L_k$ for all $\boldsymbol{s},\boldsymbol{\theta} \in \mathcal{S} \times \Theta$. The value function $V : \mathcal{S} \times \Theta \to \mathbb{R}$ has a bounded norm in the reproducing kernel Hilbert space (RKHS) defined through the kernel $k$, i.e., $\|V\|_k = \sqrt{\langle V, V \rangle} \leq B$ for some $B \in \mathbb{R}_{>0}$.

The restriction to functions with bounded RKHS norm can be commonly found in research on Bayesian optimization (Srinivas et al., 2012; Chowdhury & Gopalan, 2017) and reinforcement learning (Curi et al., 2020; Kakade et al., 2020) as it allows the analysis of learning errors. Depending on the used kernel, the RKHS can contain large function classes, e.g., analytic and Sobolev functions (van der Vaart & van Zanten, 2011; Kanagawa et al., 2018). Due to known continuity properties of value functions, e.g., (Hinderer, 2005; Harder & Peitz, 2024), this restriction is generally not severely restricting the applicability of our results.

# 3. Bayesian Optimization of Discounted Infinite-Horizon Value Functions

To iteratively select the parameters $\boldsymbol{\theta}$, we develop and analyze a novel variant of the GP-UCB algorithm that exploits structured Gaussian processes inspired by temporal difference learning. For this purpose, we firstly introduce some general background on Gaussian process regression in Section 3.1. In Section 3.2, we propose a structured GP prior that enables learning value functions from trajectory data taking rewards as training targets. Based on the resulting temporal difference GP models, we present a GP-UCB algorithm for Bayesian optimization in Section 3.3 and derive cumulative regret bounds in Section 3.4. To render these bounds fully data-dependent, we analyze the maximum information gain under our structured GP prior in Section 3.5.

## 3.1. Gaussian Process Regression

Gaussian process regression (Rasmussen & Williams, 2006) is a supervised machine learning technique that relies on Bayesian principles. A Gaussian process $g \sim \mathcal{GP}(m_g, k_g)$ can be considered a generalization of the Gaussian distribution to functions $g : \mathbb{R}^{d_x} \to \mathbb{R}$, which is fully specified through the prior mean $\mathbb{E}[g(\boldsymbol{x})] = m_g(\boldsymbol{x})$, $m_g : \mathbb{R}^{d_x} \to \mathbb{R}$, and a covariance function $\mathrm{Cov}(g(\boldsymbol{x}), g(\boldsymbol{x'})) = k_g(\boldsymbol{x}, \boldsymbol{x'})$ defined through a kernel $k_g : \mathbb{R}^{d_x} \times \mathbb{R}^{d_x} \to \mathbb{R}$. As common in literature, we set the prior mean as $m_g \equiv 0$ in the following. Given a data set $\mathbb{D} = \{(\boldsymbol{x}_n, y_n)_{n=1,\ldots,N}\}$ with training targets $y_n = g(\boldsymbol{x}_n) + \epsilon_n$ perturbed by i.i.d. Gaussian noise $\epsilon_n \sim \mathcal{N}(0, \sigma^2)$, we can formulate the joint distribution between a test output $g(\boldsymbol{x})$ and the training targets $\boldsymbol{y}$ as

$$\begin{bmatrix} \boldsymbol{y}_N \\ g(\boldsymbol{x}) \end{bmatrix} \sim \mathcal{N}\left(\begin{bmatrix} \boldsymbol{0} \\ 0 \end{bmatrix}, \begin{bmatrix} \boldsymbol{K}_g(\boldsymbol{X}_N) + \sigma^2 \boldsymbol{I} & \boldsymbol{k}_g(\boldsymbol{x}) \\ \boldsymbol{k}_g^T(\boldsymbol{x}) & k_g(\boldsymbol{x},\boldsymbol{x}) \end{bmatrix}\right), \quad (4)$$

where $[\boldsymbol{y}_N]_i = y_i$, $[\boldsymbol{X}_N]_i = \boldsymbol{x}_i$, $[\boldsymbol{K}_g(\boldsymbol{X}_N)]_{i,j} = k_g(\boldsymbol{x}_i, \boldsymbol{x}_j)$ and $[\boldsymbol{k}_g(\boldsymbol{x})]_i = k_g(\boldsymbol{x}_i, \boldsymbol{x})$ for $i, j = 1, \ldots, N$. Due to this joint distribution, the posterior conditioned on the data set is Gaussian $g(\boldsymbol{x})|\mathbb{D} \sim \mathcal{N}(\mu_g(\boldsymbol{x}), \sigma_g^2(\boldsymbol{x}))$, where

$$\mu_g(\boldsymbol{x}) = \boldsymbol{k}_g^T(\boldsymbol{x})(\boldsymbol{K}_g(\boldsymbol{X}_N) + \sigma^2 \boldsymbol{I})^{-1} \boldsymbol{y}_N, \quad (5)$$

$$\sigma_g^2(\boldsymbol{x}) = k_g(\boldsymbol{x}, \boldsymbol{x}) - \boldsymbol{k}_g^T(\boldsymbol{x})(\boldsymbol{K}_g(\boldsymbol{X}_N) + \sigma^2 \boldsymbol{I})^{-1} \boldsymbol{k}_g(\boldsymbol{x}). \quad (6)$$

The matrix $\boldsymbol{K}_g(\boldsymbol{X}_N) + \sigma^2 \boldsymbol{I}$ is not only crucial for GP regression, but also has an information-theoretic meaning as it reflects the information gain due to data. Thus, the maximum information gain (Srinivas et al., 2012)

$$\Gamma_{k_g}(N) = \max_{\boldsymbol{x}_n \in \mathcal{X}, n=1,\ldots,N} \frac{1}{2} \log \det(\boldsymbol{I} + \sigma^{-2} \boldsymbol{K}_g(\boldsymbol{X}_N)) \quad (7)$$

acts as a measure for the complexity of learning problems.

## 3.2. Learning Value Functions using Temporal Difference Gaussian Processes

Since we do not have access to measurements of $V(\boldsymbol{s}, \boldsymbol{\theta})$, we cannot directly apply GP regression to learn a model of the value function. Therefore, we take inspiration from model-free RL, where value functions are commonly learned from temporal differences (Sutton & Barto, 2018)

$$\Delta V(\boldsymbol{s}, \boldsymbol{s}^+, \boldsymbol{\theta}) = V(\boldsymbol{s}, \boldsymbol{\theta}) - \gamma V(\boldsymbol{s}^+, \boldsymbol{\theta}). \quad (8)$$

For notational simplicity, we use here the shorthand notation $\boldsymbol{s}^+ \sim p(\boldsymbol{s}^+|\boldsymbol{s}, \boldsymbol{\pi}(\boldsymbol{s}, \boldsymbol{\theta}))$. By fitting a model for the temporal difference (8), $r(\boldsymbol{s}, \boldsymbol{\pi}(\boldsymbol{s}, \boldsymbol{\theta}))$ becomes available as unbiased training target for regression because $\mathbb{E}_{\boldsymbol{s}^+}[\Delta V(\boldsymbol{s}, \boldsymbol{s}^+, \boldsymbol{\theta})] = r(\boldsymbol{s}, \boldsymbol{\pi}(\boldsymbol{s}, \boldsymbol{\theta}))$. Moreover, we can exploit the closedness of GPs under the linear operation (8) (Jidling et al., 2017; Matsumoto & Sullivan, 2024), which allows us to infer the conditional distribution of $V$ given noisy measurements of $\Delta V$ by formulating their joint distribution analogously to (4).

For this purpose, we define a structured GP prior for $\Delta V$ by assuming $V \sim \mathcal{GP}(0, k_V)$. Exploiting linearity of the expectation, we immediately obtain

$$\mathrm{Cov}(\Delta V(\boldsymbol{z}_i), \Delta V(\boldsymbol{z}_j)) \quad (9)$$
$$= k_V(\boldsymbol{x}_i, \boldsymbol{x}_j) - \gamma k_V(\boldsymbol{x}_i, \boldsymbol{x}_j^+) - \gamma k_V(\boldsymbol{x}_i^+, \boldsymbol{x}_j) + \gamma^2 k_V(\boldsymbol{x}_i^+, \boldsymbol{x}_j^+)$$

where $\boldsymbol{x} = [\boldsymbol{s}; \boldsymbol{\theta}]$, $\boldsymbol{x}^+ = [\boldsymbol{s}^+; \boldsymbol{\theta}]$, $\boldsymbol{z} = [\boldsymbol{x}; \boldsymbol{x}^+]$, and we define $V(\boldsymbol{x}) = V(\boldsymbol{s}, \boldsymbol{\theta})$ and $\Delta V(\boldsymbol{z}) = \Delta V(\boldsymbol{x}, \boldsymbol{x}^+, \boldsymbol{\theta})$ with slight abuse of notation. Thus, we obtain the structured prior $\Delta V \sim \mathcal{GP}(0, k_\Delta)$ with $k_\Delta(\boldsymbol{z}_i, \boldsymbol{z}_j) = \mathrm{Cov}(\Delta V(\boldsymbol{z}_i), \Delta V(\boldsymbol{z}_j))$. Finally, we have $\mathrm{Cov}(V(\boldsymbol{x}_i), \Delta V(\boldsymbol{z}_j)) = k_V(\boldsymbol{x}_i, \boldsymbol{x}_j) - \gamma k_V(\boldsymbol{x}_i, \boldsymbol{x}_j^+)$, such that the joint distribution between measurements with Gaussian noise, i.e., $\epsilon = \Delta V(\boldsymbol{s}, \boldsymbol{s}^+, \theta) - r(\boldsymbol{s}, \boldsymbol{\pi}(\boldsymbol{s}, \boldsymbol{\theta})) \sim$

$\mathcal{N}(0, \sigma^2)$, and the value function is given by

$$\begin{bmatrix} \boldsymbol{r} \\ V(\boldsymbol{x}) \end{bmatrix} \sim \mathcal{N}\left( \begin{bmatrix} \boldsymbol{m}_\Delta \\ m_V(\boldsymbol{x}) \end{bmatrix}, \right. \tag{10}$$

$$\left. \begin{bmatrix} \boldsymbol{K}_\Delta(\boldsymbol{Z}_N) + \sigma^2 \boldsymbol{I} & \boldsymbol{k}_V(\boldsymbol{x}) - \gamma \boldsymbol{k}_V^+(\boldsymbol{x}) \\ (\boldsymbol{k}_V(\boldsymbol{x}) - \gamma \boldsymbol{k}_V^+(\boldsymbol{x}))^T & k_V(\boldsymbol{x}, \boldsymbol{x}) \end{bmatrix} \right)$$

where $[\boldsymbol{k}_V(\boldsymbol{x})]_i = k_V(\boldsymbol{x}_i, \boldsymbol{x})$, $[\boldsymbol{k}_V^+(\boldsymbol{x})]_i = k_V(\boldsymbol{x}_i^+, \boldsymbol{x})$, $[\boldsymbol{K}_\Delta(\boldsymbol{Z}_N)]_{i,j} = k_\Delta(\boldsymbol{z}_i, \boldsymbol{z}_j)$, and $[\boldsymbol{r}]_i = r(\boldsymbol{s}_i, \boldsymbol{\pi}(\boldsymbol{s}_i, \boldsymbol{\theta}_i))$. Note that we arrange all samples in $\mathbb{D}^N$, which consists of $N$ trajectories $\tau^n$ with length $M_n$, into the sequences $\boldsymbol{x}_i$ and $\boldsymbol{x}_i^+$, $i = 1, \ldots, \sum_{n=1}^N M_n$ for notational simplicity. Based on the joint distribution (10), we can proceed analogously to Section 3.1 to obtain the posterior $V(\boldsymbol{x})|\mathbb{D}^N \sim \mathcal{N}(\mu_V(\boldsymbol{x}), \sigma_V^2(\boldsymbol{x}))$ with

$$\mu_V(\boldsymbol{x}) = (\boldsymbol{k}_V(\boldsymbol{x}) - \gamma \boldsymbol{k}_V^+(\boldsymbol{x}))^T (\boldsymbol{K}_\Delta(\boldsymbol{Z}_N) + \sigma^2 \boldsymbol{I})^{-1} \boldsymbol{r}, \tag{11}$$

$$\sigma_V^2(\boldsymbol{x}) = k_V(\boldsymbol{x}, \boldsymbol{x}) - (\boldsymbol{k}_V(\boldsymbol{x}) - \gamma \boldsymbol{k}_V^+(\boldsymbol{x}))^T$$
$$\cdot (\boldsymbol{K}_\Delta(\boldsymbol{Z}_N) + \sigma^2 \boldsymbol{I})^{-1} (\boldsymbol{k}_V(\boldsymbol{x}) - \gamma \boldsymbol{k}_V^+(\boldsymbol{x})). \tag{12}$$

Note that we assume Gaussian noise $\epsilon$ purely for the derivation of posterior GP expressions (11) and (12), but this assumption is generally not satisfied when the randomness stems from transition dynamics (1) due to nonlinearity of $V$. However, we show that the sub-Gaussianity of process noise $\boldsymbol{w}$ is preserved for sufficiently smooth value functions $V$, such that existing frequentist error bounds for GP regression are applicable (Srinivas et al., 2012; Abbasi-Yadkori, 2013; Chowdhury & Gopalan, 2017; Teutsch et al., 2026).[1]

**Proposition 3.1.** *Consider a stochastic dynamical system* (1) *satisfying Assumption* 2.1, *assume that the value function* $V$ *satisfies Assumption* 2.2. *Then, the error for learning* $V$ *from a data set* $\mathbb{D}^n$ *using Gaussian process regression with posterior mean* (11) *and variance* (12) *is bounded by*

$$|\mu_V(\boldsymbol{x}) - V(\boldsymbol{x})| \le \beta(\delta)\sigma_V(\boldsymbol{x}) \tag{13}$$

*jointly for all* $\boldsymbol{x}$ *in a compact domain* $\mathcal{S} \times \Theta$ *with probability* $1 - \delta$ *for* $\delta \in (0, 1)$, *where*

$$\beta(\delta) = \tag{14}$$
$$2\sqrt{d_s L_k} B\lambda \sqrt{\log\left(\frac{1}{\delta^2} \det\left(\boldsymbol{I} + \frac{1}{\sigma^2}\boldsymbol{K}_\Delta(\boldsymbol{Z}_N)\right)\right)} + \sigma B.$$

### 3.3. Policy Search via Bayesian Optimization

Based on this GP approach for learning value functions, we interpret policy search as a black-box optimization problem with unknown objective $J$ defined in (2). Due to the regression error bound for temporal difference learning derived in Proposition 3.1, this interpretation immediately allows the extension of upper confidence bound algorithms (Auer, 2003; Srinivas et al., 2012) to our setting. To highlight this

---

[1]Proofs for all theoretical results can be found in the appendix.

---

**Algorithm 1** TD-GP-UCB

1: Initialize $\mathbb{D}^1$ (e.g., $\mathbb{D}^1 \leftarrow \emptyset$)
2: **for** $n = 1, \ldots, N$ **do**
3:     Set $\delta_n \leftarrow \frac{6\delta}{\pi^2 n^2}$
4:     Determine $\mu_{V,n}, \sigma_{V,n}^2, \beta(\delta_n)$ using (11), (12), (31)
5:     Define $\hat{V}_n(\boldsymbol{s}, \boldsymbol{\theta}) \leftarrow \mu_{V,n}(\boldsymbol{x}) + \beta(\delta_n)\sigma_{V,n}(\boldsymbol{x})$
6:     Compute $\hat{J}_n(\boldsymbol{\theta}) = \mathbb{E}_{\boldsymbol{s}_0}[\hat{V}_n(\boldsymbol{s}_0, \boldsymbol{\theta})]$
7:     Get optimistic parameters $\boldsymbol{\theta}_n \leftarrow \arg\max_{\boldsymbol{\theta} \in \Theta} \hat{J}_n(\boldsymbol{\theta})$
8:     Roll-out $\boldsymbol{\pi}(\cdot, \boldsymbol{\theta}_n)$ for $M_n$ steps
9:     Measure $\tau^n = \{(\boldsymbol{s}_0^n, r_0^n), \ldots, (\boldsymbol{s}_{M_n}^n, r_{M_n}^n)\}$
10:    Augment data set $\mathbb{D}^{n+1} \leftarrow \mathbb{D}^n \cup \tau^n$

---

connection, we refer to the resulting algorithm, which is outlined in Algorithm 1, as **T**emporal **D**ifference **G**aussian Process **U**pper **C**onfidence **B**ound (**TD-GP-UCB**).

Starting from the value function prior, our approach computes the expectation of the upper confidence bound of our value function estimate over the initial state distribution

$$\hat{J}_n(\boldsymbol{\theta}) = \mathbb{E}_{\boldsymbol{s}_0}[\mu_{V,n}([\boldsymbol{s}_0; \boldsymbol{\theta}]) + \beta(\delta_n)\sigma_{V,n}([\boldsymbol{s}_0; \boldsymbol{\theta}])]. \tag{15}$$

Note that we select $\delta_n = 6\delta/\pi^2 n^2$ in the $n$-th episode such that the union bound guarantees the upper confidence bound to hold jointly for all $n \in \mathbb{N}$ with probability $1 - \delta$. Finally, we maximize the optimistic estimate $\hat{J}_n$ with respect to the policy parameters

$$\boldsymbol{\theta}_n = \arg\max_{\boldsymbol{\theta} \in \Theta} \hat{J}_n(\boldsymbol{\theta}) \tag{16}$$

and roll it out to get new trajectory data, such that the GP model can be updated at the start of the next episode. While the optimization problem (16) is generally non-convex, we assume to have access to an oracle which provides us with the global maximum for our theoretical analysis as common in the Bayesian optimization (Kirschner et al., 2019) and kernelized reinforcement learning literature (Vakili & Olkhovskaya, 2023) when dealing with continuous spaces.

*Remark* 3.2. The expectation over the initial state distribution in Algorithm 1 may not admit a closed-form expression, but it allows an effective sampling-based approximation.

*Remark* 3.3. Since Algorithm 1 updates the GP model with $M_n$ data points every episode, exact GP regression suffers considerably from its $\mathcal{O}((nM_n)^3)$ computational complexity. For small experiments, this complexity is still tractable in principle, but it is more convenient in practice to employ one of the many computationally efficient GP approximations, see (Liu et al., 2020) for an overview. Since the proposed TD-GP remains a GP, many of these existing approximations are readily applicable. Due to its simplicity and very low complexity, we employ a spectral feature approximation (Rahimi & Recht, 2008) in our numerical experiments, which reduces the complexity to

$\mathcal{O}(nM_n)$ in each episode $n$ and thereby fully alleviates the computational cost of GP regression.

*Remark* 3.4. The assumption of an oracle for global maxima is merely a technical requirement. It can be seen in the proof of Lemma B.1 that we can easily relax the requirement to an oracle that ouputs parameters $\hat{\boldsymbol{\theta}}_n$ with $\hat{J}_n(\boldsymbol{\theta}_n) - \hat{J}_n(\hat{\boldsymbol{\theta}}_n) \leq c$ for some $c \in \mathbb{R}_{\geq 0}$ similarly as in (Kirschner et al., 2019). If the parameter $c$ is a design choice of the oracle, we can let it shrink at the same rate as the remaining regret bound, leaving our result unaffected. This is a property that numerical optimization approaches can verifiably achieve, e.g., the simple approach of gridding, whose scalability can be improved via adaptive techniques that maintain guarantees (Jones et al., 1993). Moreover, a combination of random initializations and local optimizers can probabilistically achieve the same (Danilova et al., 2022; Huang et al., 2025).

*Remark* 3.5. When applying Algorithm 1 to high-dimensional problems, two key challenges arise. Firstly, (approximately) finding the global maximum of a non-convex high dimensional function as required in every episode of our approach is a difficult task. Numerical solvers suffer from the curse of dimensionality, such that the sub-optimality of their outputs increases or their run-time grows in practice. Secondly, global optimization explores the full parameter space, which requires learning a global model of the value function. Without structural knowledge about the value function, this learning problem is naturally hard in high dimensions. Thus, existing BO approaches for high-dimensional problems typically consider simplified problems settings, e.g., focusing on local optimality (Müller et al., 2021), assuming additional structure on the objective (Mutný & Krause, 2018), or imposing additional cost on queries (Xie et al., 2024).

## 3.4. Regret Bounds

While Algorithm 1 generally resembles standard upper confidence bound approaches in Bayesian optimization (Srinivas et al., 2012; Chowdhury & Gopalan, 2017), the theoretical analysis in these works does not immediately extend to it. The reason for this difficulty lies in the fact that the regression error bound (30) relies on the posterior variances $\sigma_V^2(\boldsymbol{x})$. The sum of these variances cannot directly be bounded through the maximum information gain analogously as in standard Bayesian optimization (Srinivas et al., 2012) since it relies on different covariance functions ($k_V$ and $k_\Delta$). We overcome this challenge by bounding the variance $\sigma_V^2(\boldsymbol{x})$ through the sum of GP variances

$$\sigma_\Delta^2(\boldsymbol{z}) = k_\Delta(\boldsymbol{z}, \boldsymbol{z}) - \boldsymbol{k}_\Delta^T(\boldsymbol{z})(\boldsymbol{K}_\Delta(\boldsymbol{Z}_N) + \sigma^2\boldsymbol{I})^{-1}\boldsymbol{k}_\Delta(\boldsymbol{z}) \quad (17)$$

along the roll-out trajectories and a residual error (c.f. Lemma B.2). Due to the discount factor, we can make the residual error arbitrarily small using trajectories with sufficiently many time steps $M_n$ in every episode $n$. By increasing this roll-out length $M_n$ with growing $n$, this enables us to obtain the following result.

**Theorem 3.6.** *Consider a dynamical system* (1) *satisfying Assumption 2.1, such that Assumption 2.2 holds for the value function $V$. Assume that parameters $\boldsymbol{\theta}^n$ are chosen via* (16) *with GP mean* (11) *and variance* (12) *learned from a data set $\mathbb{D}^n$ which is obtained from roll-outs of length $M_n$, such that $\gamma^{M_n} \leq \kappa/n^2$ for $\kappa \in (0,1)$ in each episode $n = 1, \ldots, N$. Then, the regret* (3) *satisfies*

$$R_N \leq c\sqrt{NM_N\Gamma_{k_\Delta}(N)\left(\Gamma_{k_\Delta}(NM_N) + \log(N)\right)} \quad (18)$$

*with probability $1 - \delta$ and constant $c \in \mathbb{R}_{\geq 0}$ for $N > 1$.*

While we do not explicitly state it, the constant $c$ depends on the choice of the parameter $\kappa$. However, this dependency does not have any asymptotic effect since $c$ behaves as $\mathcal{O}(1)$ for $\kappa \to 0$ due to other remaining constant dependencies. For $\kappa \to 1$, the constant $c$ behaves as $\mathcal{O}(\kappa)$, i.e., it again converges to a constant. Note that $\kappa$ neither has an asymptotic effect on $M_N$: the satisfaction of $\gamma^{M_N} \leq \kappa/N^2$ requires $M_N \geq \frac{\log(\kappa)}{\log(\gamma)} - \frac{2\log(N)}{\log(\gamma)}$, i.e., $\kappa$ essentially specifies the minimally required roll-out length. In contrast, the necessary growth rate is independently given by $\log(N)$. If we limit the growth rate of $M_N$ to this necessary one, we can simplify the bound in (18) as shown in the following.

**Corollary 3.7.** *Assume that the assumptions of Theorem 3.6 are satisfied and $M_N \in \mathcal{O}(\log(N))$ holds. Then, with probability $1 - \delta$, the regret* (3) *satisfies*

$$R_N \in \mathcal{O}^* \left(\sqrt{N}\Gamma_{k_\Delta}(N\log(N))\right) \quad (19)$$

*where $\mathcal{O}^*$ denotes asymptotic expressions up to dimension-independent logarithmic factors.*

*Remark* 3.8. Letting the roll-out length grow faster than $\mathcal{O}(\log(N))$ increases the growth rate of our regret bound. However, this is merely an artifact of our analysis which stems from the delay between GP updates and data collection. Since longer roll-outs cause a larger delay, they increase the impact of this artifact. Choosing smaller roll-out lengths that violate the requirements of Theorem 3.6 will have negative impacts since they directly affect the residual error and its asymptotics (c.f., Lemma B.4). Therefore, the performance of our method is sensitive to the roll-out length in principle, but this dependency is not critical as we can easily determine a large enough value using the residual bound $\gamma^{M_n}k_V(\boldsymbol{x}, \boldsymbol{x})$.

## 3.5. Maximum Information Gain Bounds

To finally quantify the asymptotic behavior of the regret for Algorithm 1, it remains to bound the maximum information gain $\Gamma_{k_\Delta}$. Even though bounds for many commonly used kernels (Srinivas et al., 2012; Janz et al., 2020; Vakili et al., 2021) and certain combinations of them (Krause

*Table 1.* Asymptotic regret bounds for common kernels $k_V$ with $M_N \in \mathcal{O}(\log(N))$ using information gain bounds from (Srinivas et al., 2012). Feature dimensions of linear kernels are denoted by $d$.

| linear kernel | SE kernel | Matérn kernel with param. $\nu$ |
|---|---|---|
| $\mathcal{O}^*\left(d\sqrt{N}\right)$ | $\mathcal{O}^*\left(\sqrt{N}\log(N)^{d_x+1}\right)$ | $\mathcal{O}^*\left((\sqrt{N}\log(N))^{\frac{3d_x(d_x+1)+2\nu}{2\nu+d_x(d_x+1)}}\right)$ |

& Ong, 2011) exist, the special structure of the temporal difference kernel $k_\Delta$ defined using (9) prevents the direct application of these results. Nevertheless, the temporal difference kernel $k_\Delta$ is still a composition of base kernels $k_V$, which is also reflected by the Gram matrix $\boldsymbol{K}_\Delta(\boldsymbol{Z}_N)$, i.e., the core element of the definition of the maximum information gain in (7). Taking inspiration from early work on model-free RL with GPs (Engel et al., 2003), we see that the Gram matrix $\boldsymbol{K}_\Delta(\boldsymbol{Z}_N)$ can indeed be expressed through the Gram matrix $K_V(\boldsymbol{X}_N)$ of the base kernel $k_V$ via $\boldsymbol{K}_\Delta(\boldsymbol{Z}_N) = \boldsymbol{\Xi} K_V(\tilde{\boldsymbol{X}}_N)\boldsymbol{\Xi}^T$. Here, the matrix $\boldsymbol{\Xi} = \mathrm{blkdiag}(\boldsymbol{\Xi}_1, \ldots, \boldsymbol{\Xi}_N)$ with blocks

$$\boldsymbol{\Xi}_n = \begin{bmatrix} \boldsymbol{I} & \boldsymbol{0}_{M_n \times 1} \end{bmatrix} - \begin{bmatrix} \boldsymbol{0}_{M_n \times 1} & \mathrm{diag}([\gamma, \ldots, \gamma]) \end{bmatrix} \quad (20)$$

encodes the temporal difference structure of the data with each block corresponding to one roll-out trajectory. By exploiting bounds for the eigenvalues of tri-diagonal Toeplitz matrices (Kulkarni et al., 1999), we show that the norm of these matrices satisfies $\|\boldsymbol{\Xi}\| \leq (1+\gamma)$. This property essentially allows us to lump the effect of the temporal difference structure into the noise parameter in (7), such that bounds for the maximum information gain $\Gamma_{k_\Delta}$ asymptotically behave the same as $\Gamma_{k_V}$ yielding the following result.

**Theorem 3.9.** *The maximum information gain for the temporal difference kernel $k_\Delta$ is asymptotically bounded by the information gain of its base kernel, i.e.,*

$$\Gamma_{k_\Delta}(NM_N) \in \mathcal{O}(\Gamma_{k_V}(NM_N)) \quad (21)$$

*for a non-decreasing sequence $M_n$, $n = 1, \ldots, N$.*

By combining this result with Corollary 3.7, we immediately obtain the regret bound $\mathcal{O}^*(N^{1/2}\Gamma_{k_V}(N\log(N)))$. This allows us to formulate explicit regret bounds for many frequently used base kernels $k_V$ as illustrated in Table 1. Moreover, it enables a straightforward comparison with existing approaches that consider related problems, but whose guarantees generally do not extend to our setting. For example, the original GP-UCB algorithm ensures $\mathcal{O}^*(N^{1/2}\Gamma_{k_V}(N))$ (Srinivas et al., 2012), while finite-horizon kernelized reinforcement learning achieves $\mathcal{O}^*(MN^{1/2}\Gamma_{k_V}(N))$ (Chowdhury & Oliveira, 2023) with $M$ denoting the planning horizon. Hence, our asymptotic regret bound is worse by a mere $\log(N)$ term, which is likely to be an artifact of the delayed update of the GP model as discussed in Remark 3.8. This artifact can be similarly observed in model-based RL formulations (Curi

et al., 2020). For the special case of linear kernels, a regret of $\mathcal{O}^*(d\sqrt{N})$ can be achieved in a *non-episodic* infinite-horizon discounted problem (Zhou et al., 2021), which matches our result. These comparisons clearly underline the efficiency of the proposed TD-GP-UCB algorithm.

# 4. Related Work

Numerous variants of Bayesian optimization have been proposed for policy search. Many works focus on the parameterization of policies, among which LQR policies are probably the most widely investigated. Developed approaches range from the optimization over controller parameters (Calandra et al., 2015), LQR weight matrices (Marco et al., 2016), dynamics parameters for LQR (Bansal et al., 2017), and combinations thereof (Fröhlich et al., 2019). In addition to the parameterization, approaches have been tailored to specific control architectures (Khosravi et al., 2022), and properties of GPs have been exploited to combine simulation and real-world data, such that the data efficiency of policy search can be improved (Rai et al., 2019; Letham & Bakshy, 2019; He et al., 2025). However, it has only been noticed recently that the sequential structure of policy search problems can be incorporated into the prior GP mean function to reduce the sample complexity (Kallel et al., 2024).

The general idea of combining temporal difference learning with GP models to infer discounted infinite-horizon value functions originates from the field of Bayesian RL (Engel et al., 2003; Ghavamzadeh et al., 2015). It has been employed in a variety of GP-based algorithms including SARSA (Engel et al., 2005), approximate policy iteration schemes (Bethke & How, 2009; Lederer & Hirche, 2019), and Q-learning (Grande et al., 2014; Chowdhary et al., 2014), but no theoretical results comparable to ours have been derived. Independently from these Bayesian approaches, similar ideas have been exploited in least squares temporal difference learning (Bradtke & Barto, 1996), which is commonly exploited for approximate policy iteration (Lagoudakis & Parr, 2003). Convergence results can be shown for variants of this least squares policy iteration (Antos et al., 2008; Lazaric et al., 2012; Tu & Recht, 2018), which can also be extended to regularized versions (Farahmand et al., 2016). However, these regularized approaches lead to model structures which do not allow a straightforward adaptation to GP regression, despite the general connection between these techniques (Kanagawa et al., 2018).

In contrast to Bayesian RL and least squares policy iteration, upper confidence bounds play a crucial role in many approaches for value iteration using linear function approximation. These methods do not directly determine value functions in closed-form using TD learning, but iteratively construct them using optimistic least squares iterations (Jin et al., 2020). This approach is particularly well suited for

episodic finite horizon planning problems with finite action spaces, such that regret bounds for multiple algorithms resembling Q-learning have been derived for (kernelized) linear MDPs (Yang et al., 2020; Yang & Wang, 2020; Chowdhury & Oliveira, 2023). These approaches can be extended to discounted infinite horizon problems by changing to non-episodic, online learning algorithms (Zhou et al., 2021; Chen et al., 2022). Moreover, uncertainty sampling can be employed to find $\epsilon$-optimal policies when the transition probability is in the RKHS of the used kernel (Yeh et al., 2023). While such approximate value iteration approaches can provide strong theoretical guarantees, value iteration requires the computation of optimal actions. For continuous action spaces, this implies solving at least one generally non-convex optimization problem per data point in every episode, which severely impacts the practicality of these algorithms.

Note that the practical effectiveness of learning the value function in dependency of the policy parameters has been demonstrated experimentally. State- and parameter-dependent value functions are learned using deep RL in (Faccio et al., 2021). Moreover, policy evaluation networks only depending on a 'fingerprint' of a policy neural network are successfully employed in (Harb et al., 2020). These examples further support our rationale of learning policy parameter-dependent value functions.

## 5. Numerical Evaluation

We demonstrate the effectiveness of the proposed TD-GP-UCB algorithm[2] in modified benchmark problems from the Gymnasium Classic Control Suite (Towers et al., 2024) and an adapted LQR design problem (Dean et al., 2020). In Section 5.1, we describe the setting that is used for evaluation and we outline the baselines that we compare against. We present the results for Gymnasium experiments with dense rewards in Section 5.2 and for experiments with sparse rewards in Section 5.3. Finally, we illustrate the effectiveness of our approach for the adapted LQR problem in Section 5.4.

### 5.1. Simulation Setting and Baselines

As linear policies have been demonstrated to achieve a competitive performance on many RL benchmarks (Mania et al., 2018), we also adopt this policy class by choosing $\pi(s, \theta) = \theta^\top s$, with bounded parameters $\theta \in [-1, 1]^{d_s}$. Following the discussion in Remark 3.8, we apply these policies for $\lceil \frac{10}{1-\gamma} + \log(n) \rceil$ time steps in episode $n$ of every environment, such that $\gamma^{M_n} \leq e^{-10} \leq 10^{-4}$. We determine returns using $\gamma = 0.9$ and evaluate the performance of all algorithms using the cumulative regret averaged over 20 runs with different random seeds. To compute the

[2]Our implementation is publicly available at https://github.com/anuj27596/tdgp.

regret, we find the optimal parameters $\theta^*$ by numerically maximizing the returns using the Nelder-Mead method.

Since each episode generates more than 100 training samples for the temporal difference GP in Algorithm 1, data quickly accumulates. Therefore, we use an approximation of the squared exponential kernel based on 200 random spectral features (Rahimi & Recht, 2008) as our base kernel $k_V$. Unless stated differently, we start each run with the roll-out of a random parameter $\theta$. The hyperparameters of the approximated squared exponential kernel are tuned using log-likelihood maximization after 10 and 20 subsequent episodes, respectively. As common in the Bayesian optimization literature (Fiedler et al., 2021), we fix $\beta(\delta) = 2$. Moreover, we approximate the expectation in (15) via the empirical mean over 10 samples that are fixed at the beginning of each run. The maximization in (16) is executed using the Nelder-Mead method.

We compare our TD-GP-UCB algorithm to three baselines:

- **MC-GP-UCB:** To demonstrate the benefits of exploiting the temporal difference decomposition of value functions in the GP model, we compare to most common variant of GP-UCB that directly uses the Monte-Carlo value function estimates as training targets.

- **kQ-learn:** As an example for a variant of kernelized Q-learning, we adapt the approach for optimistic value iteration in (Chowdhury & Oliveira, 2023) to our problem setting. To avoid excessive computation times, we limit the number of value iterations to 10 per episode.

- **DDPG:** To illustrate benefits over modern reinforcement learning techniques, we use Deep Deterministic Policy Gradients (Lillicrap et al., 2016) as an example for the ubiquitous class of actor-critic methods. We constrain the actor to linear policies with the same parameter bounds as our method for a fair comparison.

### 5.2. Dense Reward Experiments

We firstly compare all methods on the Gymnasium Pendulum environment using the default squared rewards, which provide dense feedback. We slightly modify the environment by defining the angular position and angular velocity as observations, such that the linear controller parameterization is essentially rendered a PD control law (Fadali & Visioli, 2012). We evaluate all methods based on three different distributions of the initial state: **random**, **downwards**, and **upwards**. When evaluating on the **random** scenario, which corresponds to the default setting of the environment, the initial angular position and velocity are sampled uniformly at random from the entire state space. Thereby, the variance of episode returns is naturally large in this scenario. In order to reduce the randomness due to the initial state distribution, we additionally evaluate the methods in the **downwards** and **upwards** scenarios, where

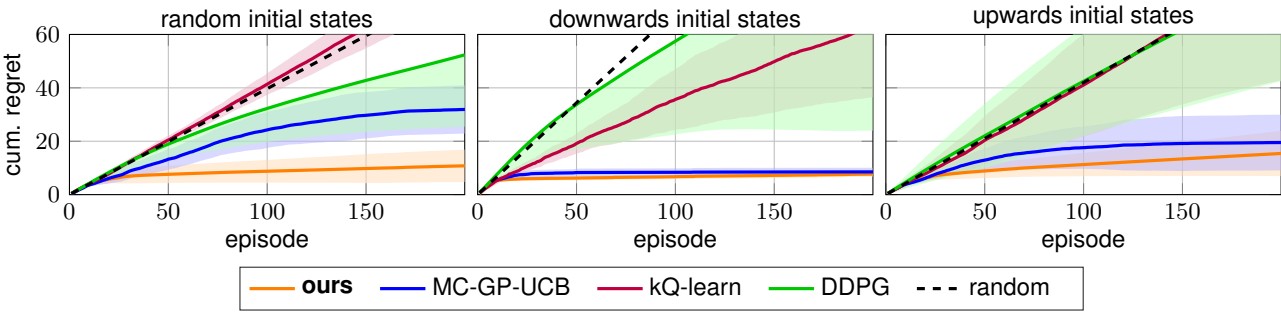

*Figure 2.* Average cumulative regret for the Gymnasium Pendulum environment with dense rewards for three different initial state distributions. Shaded areas illustrate one standard deviation confidence intervals.

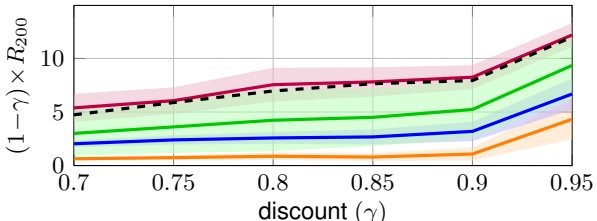

*Figure 3.* Behavior of cumulative regret normalized by effective horizon depending on the discount factor.

the initial state is sampled from uniform distributions with width 0.1 centered around the downwards and upwards stationary configurations, respectively. These settings are highly distinctive from each other: Due to the quadratic rewards, early rewards of roll-outs are relatively flat and optimal returns have small magnitudes in the **upwards** scenario, while the opposite holds for the **downwards** scenario.

The variety of these scenarios highlights crucial differences in the regret of the compared methods as illustrated in Figure 2. DDPG only achieves a performance improvement over random actions in the scenario with random initial states. Since it does not benefit from the exploration provided by the initial states in other scenarios, more than 200 episodes are necessary before the regret decays. A similar behavior is visible for kQ-learn with an improvement merely for downwards initial states, where early rewards provide strong signals for performance improvement. MC-GP-UCB suffers from the high variance of episode returns in the random scenario. While it also struggles with flat rewards in the early steps of roll-outs experienced in the upwards scenario, our proposed TD-GP-UCB approach shows a consistent performance across all scenarios.

While it would seem reasonable that the advantage of exploiting the temporal difference structure becomes less beneficial with shorter effective horizons, i.e., smaller $\gamma$, this is generally not the case. As illustrated in Figure 3 for the random scenario, the cumulative regret of our TD-GP-UCB approach normalized by effective horizon lies consistently

below all baseline approaches. It only exhibits a considerable increase when $\gamma$ is close to 1, which coincides with the performance deterioration of random policies. Hence, our method's regret strongly correlates with the overall problem difficulty, which further underlines its effectiveness.

### 5.3. Sparse Reward Experiments

To increase the problem difficulty, we additionally modify the scenarios considered in Section 5.2 by providing only binary rewards, such that we obtain a sparse reward setting. In particular, a reward of 1 is returned if the angular position is within 60 degrees of the vertically upward position and 0 otherwise. Note that the binary rewards in combination with the deterministic pendulum dynamics render the value functions discontinuous. To account for this violation of our theoretical assumptions, we set $\beta(\delta) = 2 + 0.1\sqrt{n}$ following (Bogunovic & Krause, 2021).

The cumulative regret achieved in the modified random and upwards scenarios is illustrated at the left and center of Figure 4. Even though the proposed TD-GP-UCB algorithm remains to outperform the baseline methods, a considerable increase of the regret can be observed for all methods except DDPG. In particular, the performance of our approach suffers significantly from the high randomness in the random scenario. It should be noted that the apparently linear growth of the cumulative regret is not out of the ordinary: It is known that Bayesian optimization exhibits such a regret in the misspecified setting (Bogunovic & Krause, 2021) to which these scenarios correspond to.

While our approach relies on the smoothness of value functions, we can further violate this condition by considering discrete actions. We demonstrate this for the Gymnasium Cart-Pole environment that admits only actions $\{0, 1\}$. We filter the output of the linear policy using the Heaviside step function to enable our approach. Further, in the case of DDPG we use the straight-through estimator for backpropagation. To account for the higher-dimensional state space, we additionally collect 5 initial rollouts using random parameters $\boldsymbol{\theta}$, but leave the evaluation setting otherwise unchanged

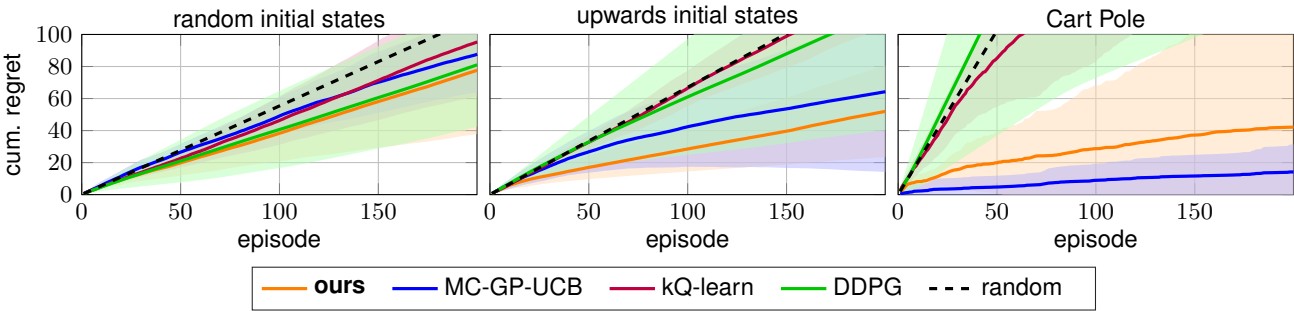

*Figure 4.* Average cumulative regret for the Gymnasium Pendulum and Cart Pole environments with sparse rewards for different initial state distributions. Shaded areas illustrate one standard deviation confidence intervals.

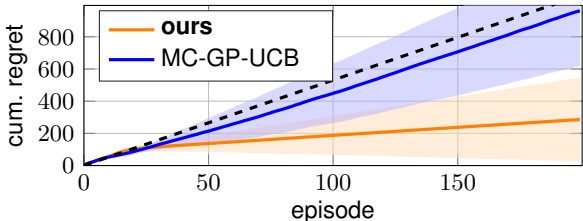

*Figure 5.* Average cumulative regret for the adapted LQR problem with unstable open-loop dynamics based on (Dean et al., 2020).

compared to the Pendulum environment with dense rewards. The resulting regret curves are illustrated on the right of Figure 4. While the performance of our approach is worse than that of the default GP-UCB algorithm, this is mainly caused by a steep regret growth before the first hyperparemeter optimization, while the regret grows very slowly afterwards. Therefore, these examples demonstrate the practical effectiveness of our approach beyond its theoretical guarantees.

### 5.4. LQR Control Design

Finally, we benchmark our approach in a challenging control design example adapted from the LQR problem proposed in (Dean et al., 2020). To align with our setting, we consider a discounted version and squash rewards through a transform $r' = \tanh(\frac{r}{200})$, such that the value function exists despite noisy dynamics. For TD-GP-UCB, we additionally squash the state training inputs via the sigmoid function $s'_i = \frac{1000}{1+e^{-s_i/1000}} - 500$ to deal with divergence caused by the open-loop instability of the dynamics. Design choices concerning the policy search methods are identical as in Section 5.2. Similarly to the results for the Inverted Pendulum environment with dense rewards, we find that our approach generally outperforms standard Bayesian optimization using GP models with Monte-Carlo value function targets. In particular, MC-GP-UCB achieves only a slight improvement over the random baseline for the adapted LQR problem within the executed number of episodes, while the proposed TD-GP-UCB algorithm significantly improves upon it. This confirms the strength of our proposed method.

## 6. Conclusion and Outlook

In this work, we proposed a novel framework for temporal difference-based learning of value functions with Gaussian process regression. We established regret bounds for UCB-based BO in this framework, for which we analyzed the maximum information gain of the temporal difference GP. The effectiveness of our approach is demonstrated in several benchmarks additionally highlighting its practical benefits.

In future work, we aim to improve the scalability of our method to high dimensional state and parameter spaces. Following the discussion in Remark 3.5, a reduction of the sample complexity of GP learning via informative task-specific priors (Rai et al., 2019; Rothfuss et al., 2023) seems a promising direction. These approaches can be combined with a dimensionality reduction of the policy parameterization, e.g., via so call fingerprints of neural network policies (Harb et al., 2020). Moreover, we can adapt our approach to focus on local optimality along the lines of (Müller et al., 2021; Nguyen et al., 2022), such that local model accuracy suffices. With such modifications, we believe that our approach can be rendered applicable to higher dimensional problems such as certifiable neural network policy fine-tuning.

## Acknowledgements

We acknowledge the support from the Swiss National Science Foundation under NCCR Automation, grant agreement 51NF40 180545, and a start-up grant of the National University of Singapore. Marco Bagatella is supported by the Max Planck ETH Center for Learning Systems.

## Impact Statement

This paper presents work whose goal is to advance the field of Machine Learning. There are many potential societal consequences of our work, none which we feel must be specifically highlighted here.

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

# A. Error bounds for Gaussian process regression

**Lemma A.1.** *Consider a value function $V$ satisfying Assumption 2.2. Then, $V$ is $L_V$-Lipschitz continuous with $L_V = \sqrt{L_k}B$.*

*Proof.* It follows from (Steinwart & Christmann, 2008, Corollary 4.36) that a function with bounded RKHS norm satisfies

$$\left| \frac{\partial}{\partial s_i} V(\boldsymbol{s}, \boldsymbol{\theta}) \right|^2 \leq B^2 \frac{\partial^2}{\partial s_i \partial s_i'} k([\boldsymbol{s}; \boldsymbol{\theta}], [\boldsymbol{s}'; \boldsymbol{\theta}]) \Big|_{\boldsymbol{s}'=\boldsymbol{s}}. \tag{22}$$

By defining the Lipschitz constant via the maximum derivative of $V$, we consequently obtain

$$L_V \leq \max_{\boldsymbol{s}\in\mathcal{S}, \boldsymbol{\theta}\in\Theta} \|\nabla_{\boldsymbol{s}} V(\boldsymbol{s}, \boldsymbol{\theta})\| \tag{23}$$

$$\leq \max_{\boldsymbol{s}\in\mathcal{S}, \boldsymbol{\theta}\in\Theta} \left\| \begin{bmatrix} \frac{\partial}{\partial s_1} V(\boldsymbol{s}, \boldsymbol{\theta}) \\ \vdots \\ \frac{\partial}{\partial s_{d_s}} V(\boldsymbol{s}, \boldsymbol{\theta}) \end{bmatrix} \right\| \tag{24}$$

$$\leq B \max_{\boldsymbol{s}\in\mathcal{S}, \boldsymbol{\theta}\in\Theta} \left\| \begin{bmatrix} \sqrt{\frac{\partial^2}{\partial s_1 \partial s_1'} k([\boldsymbol{s}; \boldsymbol{\theta}], [\boldsymbol{s}'; \boldsymbol{\theta}])\big|_{\boldsymbol{s}'=\boldsymbol{s}}} \\ \vdots \\ \sqrt{\frac{\partial^2}{\partial s_{d_s} \partial s_{d_s}'} k([\boldsymbol{s}; \boldsymbol{\theta}], [\boldsymbol{s}'; \boldsymbol{\theta}])\big|_{\boldsymbol{s}'=\boldsymbol{s}}} \end{bmatrix} \right\| \tag{25}$$

$$\leq B \max_{\boldsymbol{s}\in\mathcal{S}, \boldsymbol{\theta}\in\Theta} \sqrt{\sum_{i=1}^{d_s} \frac{\partial^2}{\partial s_i \partial s_i'} k([\boldsymbol{s}; \boldsymbol{\theta}], [\boldsymbol{s}'; \boldsymbol{\theta}])\Big|_{\boldsymbol{s}'=\boldsymbol{s}}} \tag{26}$$

$$\leq B \max_{\boldsymbol{s}\in\mathcal{S}, \boldsymbol{\theta}\in\Theta} \sqrt{\operatorname{trace}\left( \nabla_{\boldsymbol{s}} \nabla_{\boldsymbol{s}'}^T k([\boldsymbol{s}; \boldsymbol{\theta}], [\boldsymbol{s}'; \boldsymbol{\theta}])\Big|_{\boldsymbol{s}'=\boldsymbol{s}} \right)}. \tag{27}$$

Finally, the result follows by bounding the trace using $L_k$. $\square$

**Lemma A.2.** *Consider a $L_V$-Lipschitz value function $V$ and process noise satisfying Assumption 2.1. Then, $\Delta V(\boldsymbol{s}, \boldsymbol{s}^+, \boldsymbol{\theta}) - r(\boldsymbol{s}, \boldsymbol{\pi}(\boldsymbol{s}, \boldsymbol{\theta}))$ is $2\sqrt{d_s}L_V\lambda$-sub-Gaussian.*

*Proof.* It directly follows from the definition of $\Delta V$ that

$$\mathbb{E}_{\boldsymbol{w}}[\Delta V(\boldsymbol{s}, \boldsymbol{s}^+, \boldsymbol{\theta})] = r(\boldsymbol{s}, \boldsymbol{\pi}(\boldsymbol{s}, \boldsymbol{\theta})).$$

Moreover, it is straightforward to see that all the randomness of $\Delta V(\boldsymbol{s}, \boldsymbol{s}^+, \boldsymbol{\theta})$ appears in

$$V(\boldsymbol{s}^+, \boldsymbol{\theta}) = V(\tilde{\boldsymbol{s}}^+ + \boldsymbol{w}, \boldsymbol{\theta})$$

with $\tilde{\boldsymbol{s}}^+ = \mathbb{E}_{\boldsymbol{s}^+}[\boldsymbol{s}^+]$ for $\boldsymbol{s}^+ \sim p(\boldsymbol{s}^+|\boldsymbol{s}, \boldsymbol{\pi}(\boldsymbol{s}, \boldsymbol{\theta}))$ and $\boldsymbol{w} = \boldsymbol{s}^+ - \tilde{\boldsymbol{s}}^+$. Therefore, it suffices to show

$$\mathbb{E}_{\boldsymbol{w}}\left[ \exp\left( \frac{(V(\tilde{\boldsymbol{s}}^+ + \boldsymbol{w}, \boldsymbol{\theta}) - \mathbb{E}_{\boldsymbol{w}}[V(\tilde{\boldsymbol{s}}^+ + \boldsymbol{w}, \boldsymbol{\theta})])^2}{4 d_s L_V^2 \lambda^2} \right) \right] \leq 2 \tag{28}$$

to prove that $\Delta V(\boldsymbol{s}, \boldsymbol{s}^+, \boldsymbol{\theta}) - r(\boldsymbol{s}, \boldsymbol{\pi}(\boldsymbol{s}, \boldsymbol{\theta}))$ is $2\sqrt{d_s}L_V\lambda$-sub-Gaussian. For this purpose, we let $\boldsymbol{w}'$ be an independent copy of $\boldsymbol{w}$ similarly as in (Begzadić et al., 2025). Since $\boldsymbol{w}'$ has the same distribution as $\boldsymbol{w}$, we can reformulate the left side of (28) as

$$\mathbb{E}_{\boldsymbol{w}}\left[ \exp\left( \frac{(V(\tilde{\boldsymbol{s}}^+ + \boldsymbol{w}, \boldsymbol{\theta}) - \mathbb{E}_{\boldsymbol{w}}[V(\tilde{\boldsymbol{s}}^+ + \boldsymbol{w}, \boldsymbol{\theta})])^2}{4 d_s L_V^2 \lambda^2} \right) \right] = \\ \mathbb{E}_{\boldsymbol{w}}\left[ \exp\left( \frac{(V(\tilde{\boldsymbol{s}}^+ + \boldsymbol{w}, \boldsymbol{\theta}) - \mathbb{E}_{\boldsymbol{w}'}[V(\tilde{\boldsymbol{s}}^+ + \boldsymbol{w}', \boldsymbol{\theta})])^2}{4 d_s L_V^2 \lambda^2} \right) \right].$$

Using Jensen's inequality, we can pull the expectation out of the exponential yielding

$$\exp\left(\frac{(V(\tilde{\boldsymbol{s}}^+ + \boldsymbol{w}, \boldsymbol{\theta}) - \mathbb{E}_{\boldsymbol{w}}[V(\tilde{\boldsymbol{s}}^+ + \boldsymbol{w}, \boldsymbol{\theta})])^2}{4d_s L_V^2 \lambda^2}\right) \leq$$
$$\mathbb{E}_{\boldsymbol{w}'}\left[\exp\left(\frac{(V(\tilde{\boldsymbol{s}}^+ + \boldsymbol{w}, \boldsymbol{\theta}) - V(\tilde{\boldsymbol{s}}^+ + \boldsymbol{w}', \boldsymbol{\theta}))^2}{4d_s L_V^2 \lambda^2}\right)\right].$$

This allows us to exploit the Lipschitz continuity of $V$ via

$$\mathbb{E}_{\boldsymbol{w},\boldsymbol{w}'}\left[\exp\left(\frac{(V(\tilde{\boldsymbol{s}}^+ + \boldsymbol{w}, \boldsymbol{\theta}) - V(\tilde{\boldsymbol{s}}^+ + \boldsymbol{w}', \boldsymbol{\theta}))^2}{4d_s L_V^2 \lambda^2}\right)\right] \leq$$
$$\mathbb{E}_{\boldsymbol{w},\boldsymbol{w}'}\left[\exp\left(\frac{(L_V(\|\boldsymbol{w}\| + \|\boldsymbol{w}'\|))^2}{4d_s L_V^2 \lambda^2}\right)\right].$$

Due to Young's inequality we can bound this expression in terms of $\|\boldsymbol{w}\|^2$ and $\|\boldsymbol{w}'\|^2$, i.e.,

$$\mathbb{E}_{\boldsymbol{w},\boldsymbol{w}'}\left[\exp\left(\frac{(L_V(\|\boldsymbol{w}\| + \|\boldsymbol{w}'\|))^2}{4d_s L_V^2 \lambda^2}\right)\right] \leq \mathbb{E}_{\boldsymbol{w},\boldsymbol{w}'}\left[\exp\left(\frac{\|\boldsymbol{w}\|^2 + \|\boldsymbol{w}'\|^2}{2d_s \lambda^2}\right)\right].$$

The independence of $\boldsymbol{w}$ and $\boldsymbol{w}'$ and their identical distributions ensures that

$$\mathbb{E}_{\boldsymbol{w},\boldsymbol{w}'}\left[\exp\left(\frac{\|\boldsymbol{w}\|^2 + \|\boldsymbol{w}'\|^2}{2d_s \lambda^2}\right)\right] = \mathbb{E}_{\boldsymbol{w},\boldsymbol{w}'}\left[\exp\left(\frac{\|\boldsymbol{w}\|^2}{2d_s \lambda^2}\right)\right]^2$$
$$\leq \mathbb{E}_{\boldsymbol{w}}\left[\exp\left(\frac{\|\boldsymbol{w}\|^2}{d_s \lambda^2}\right)\right],$$

where the second line follows from Jensen's inequality. Finally, observe that $\|\boldsymbol{w}\|^2 = \sum_{i=1}^{d_s} w_i^2 = \sum_{i=1}^{d_s} (\boldsymbol{v}_i^T \boldsymbol{w})^2$, where $\boldsymbol{v}_1, \ldots, \boldsymbol{v}_{d_s}$ is the standard orthonormal basis of $\mathbb{R}^{d_s}$. Therefore, we obtain

$$\mathbb{E}_{\boldsymbol{w}}\left[\exp\left(\frac{\|\boldsymbol{w}\|^2}{d_s \lambda^2}\right)\right] \leq \mathbb{E}_{\boldsymbol{w}}\left[\prod_{i=1}^{d_s} \exp\left(\frac{w_i^2}{d_s \lambda^2}\right)\right]$$
$$\leq \prod_{i=1}^{d_s} \mathbb{E}_{\boldsymbol{w}}\left[\exp\left(\frac{w_i^2}{\lambda^2}\right)\right]^{\frac{1}{d_s}}$$
$$\leq \prod_{i=1}^{d_s} \mathbb{E}_{\boldsymbol{w}}\left[\exp\left(\frac{(\boldsymbol{v}_i^T \boldsymbol{w})^2}{\lambda^2}\right)\right]^{\frac{1}{d_s}},$$

where the second line follows from the Cauchy-Schwarz inequality. Since $\boldsymbol{v}^T \boldsymbol{w}$ is $\lambda$-sub-Gaussian for every $\boldsymbol{v}$ due to Assumption 2.1, we have

$$\prod_{i=1}^{d_s} \mathbb{E}_{\boldsymbol{w}}\left[\exp\left(\frac{(\boldsymbol{v}_i^T \boldsymbol{w})^2}{\lambda^2}\right)\right]^{\frac{1}{d_s}} \leq \prod_{i=1}^{d_s} 2^{\frac{1}{d_s}} = 2, \tag{29}$$

which concludes the proof.

$\square$

**Proposition 3.1.** *Consider a stochastic dynamical system* (1) *satisfying Assumption 2.1, assume that the value function $V$ satisfies Assumption 2.2, and let $m_V \equiv 0$. Then, the error for learning $V$ from a data set $\mathbb{D}^n$ using Gaussian process regression with posterior mean* (11) *and variance* (12) *is bounded by*

$$|\mu_V(\boldsymbol{x}) - V(\boldsymbol{x})| \leq \beta(\delta)\sigma_V(\boldsymbol{x}) \tag{30}$$

*jointly for all $\boldsymbol{x}, \boldsymbol{\theta}$ in a compact domain $\mathcal{S} \times \Theta$ with probability $1 - \delta$, $\delta \in (0, 1)$, where*

$$\beta(\delta) = 2\sqrt{d_s L_k} B\lambda \sqrt{\log \det\left(\boldsymbol{I} + \frac{1}{\sigma^2} \boldsymbol{K}_\Delta(\boldsymbol{Z}_N)\right) - 2\log(\delta)} + \sigma B. \tag{31}$$

*Proof.* The target values for the GP model with prior mean $m \equiv 0$ and kernel (9) are of the form $\Delta V(s, s^+, \boldsymbol{\theta})$ but we use labels $r(s, \boldsymbol{\pi}(s, \boldsymbol{\theta}))$ for computing the posterior mean (11). Thus, the labels are perturbed by noise $\Delta V(s, s^+, \boldsymbol{\theta}) - r(s, \boldsymbol{\pi}(s, \boldsymbol{\theta}))$. Note that the value function $V$ is $L_V$-Lipschitz with $L_V = \sqrt{L_k}B$ due to Lemma A.1. Therefore, it follows from Lemma A.2 that $\Delta V(s, s^+, \boldsymbol{\theta}) - r(s, \boldsymbol{\pi}(s, \boldsymbol{\theta}))$ is $2\sqrt{d_s L_k}B\lambda$-sub-Gaussian. Finally, the sub-Gaussianity of the noise in GP regression allows the application of (Abbasi-Yadkori, 2013, Theorem 3.11), which guarantees a learning error bound (30) with $\beta$ defined in (31). $\qquad\square$

## B. Regret Bounds for Optimistic Optimization

**Lemma B.1.** *Consider a dynamical system* (1) *with process noise satisfying Assumption 2.1 and assume that the value function $V$ satisfies Assumption 2.2. If $\boldsymbol{\theta}_{\mathrm{UCB}}$ is chosen according to* (16) *it holds with probability $1 - \delta$ that*

$$J(\boldsymbol{\theta}^*) - J(\boldsymbol{\theta}_{\mathrm{UCB}}) \leq 2\beta \mathbb{E}_{\boldsymbol{s}_0}\left[\sigma_V\left([\boldsymbol{s}_0; \boldsymbol{\theta}_{\mathrm{UCB}}]\right)\right]. \tag{32}$$

*Proof.* Due to Proposition 3.1, $\hat{J}$ upper bounds $J$ with probability $1 - \delta$. Thus, we have

$$J(\boldsymbol{\theta}^*) \leq \hat{J}(\boldsymbol{\theta}^*)$$

with probability $1 - \delta$. Moreover, (16) implies that

$$\hat{J}(\boldsymbol{\theta}^*) \leq \hat{J}(\boldsymbol{\theta}_{\mathrm{UCB}}).$$

Combining these two inequalities, we obtain

$$J(\boldsymbol{\theta}^*) - J(\boldsymbol{\theta}_{\mathrm{UCB}}) \leq \hat{J}(\boldsymbol{\theta}_{\mathrm{UCB}}) - J(\boldsymbol{\theta}_{\mathrm{UCB}})$$

with probability $1 - \delta$. Exploiting the linearity of the expectation and the definition of $\hat{J}$ yields

$$J(\boldsymbol{\theta}^*) - J(\boldsymbol{\theta}_{\mathrm{UCB}}) \leq \mathbb{E}_{\boldsymbol{s}_0}\left[\mu_V\left([\boldsymbol{s}_0; \boldsymbol{\theta}_{\mathrm{UCB}}]\right) + \beta\sigma_V\left([\boldsymbol{s}_0; \boldsymbol{\theta}_{\mathrm{UCB}}]\right) - V(\boldsymbol{s}_0, \boldsymbol{\theta}_{\mathrm{UCB}})\right]$$

with probability $1 - \delta$. Finally, we can bound $|\mu_V\left([\boldsymbol{s}_0; \boldsymbol{\theta}_{\mathrm{UCB}}]\right) - V(\boldsymbol{s}_0, \boldsymbol{\theta}_{\mathrm{UCB}})| \leq \beta^n \sigma_V\left([\boldsymbol{s}_0; \boldsymbol{\theta}_{\mathrm{UCB}}]\right)$ with probability $1 - \delta$ by employing Proposition 3.1 once more, which immediately implies (32).

$\qquad\square$

**Lemma B.2.** *Let $\tau^n = \{(\boldsymbol{s}_m, r_m)_{m=1,\ldots,M}\}$, $\boldsymbol{x}_m = [\boldsymbol{s}_m, \boldsymbol{\theta}]$, and $\boldsymbol{z}_m = [\boldsymbol{x}_m, \boldsymbol{x}_{m+1}]$. Then, the posterior variance* (12) *satisfies*

$$\sigma_V^2(\boldsymbol{x}_0) \leq (M+1)\gamma^M \sigma_V^2(\boldsymbol{x}_M) + \frac{\gamma^M - \gamma^{M+1} + 1}{1 - \gamma} \sum_{m=0}^{M-1} \gamma^m \sigma_\Delta^2(\boldsymbol{z}_m) \tag{33}$$

*for $\sigma_\Delta^2$ defined in* (17).

*Proof.* Due to the Bayesian foundations of GPs, we know that $\sigma_V^2(\boldsymbol{x}_0)$ is the conditional variance of $V(\boldsymbol{x}_0)$ under the GP prior, i.e.,

$$\sigma_V^2(\boldsymbol{x}_0) = \mathbb{V}\left[V(\boldsymbol{x}_0)|\mathbb{D}\right]. \tag{34}$$

For notational simplicity, we drop the conditioning on data for notational simplicity in the following derivations and simply write $\sigma_V^2(\boldsymbol{x}_0) = \mathbb{V}\left[V(\boldsymbol{x}_0)\right]$. Observe that we can expand the value function in terms of the temporal difference via

$$\begin{aligned} V(\boldsymbol{x}_m) &= V(\boldsymbol{x}_m) - \gamma V(\boldsymbol{x}_{m+1}) + \gamma V(\boldsymbol{x}_{m+1}), \\ &= \Delta V(\boldsymbol{z}_m) + \gamma V(\boldsymbol{x}_{m+1}). \end{aligned}$$

Employing this identity recursively, we obtain

$$V(\boldsymbol{x}_0) = \gamma^M V(\boldsymbol{x}_M) + \sum_{m=0}^{M-1} \gamma^m \Delta V(\boldsymbol{z}_m).$$

Therefore, we can express the GP variance $\sigma_V^2(\boldsymbol{x}_0)$ as

$$\sigma_V^2(\boldsymbol{x}_0) = \mathbb{V}\left[\gamma^M V(\boldsymbol{x}_M) + \sum_{m=0}^{M-1} \gamma^m \Delta V(\boldsymbol{z}_m)\right],$$

which we can equivalently express as

$$\sigma_V^2(\boldsymbol{x}_0) = \text{Cov}\left(\gamma^M V(\boldsymbol{x}_M), \gamma^M V(\boldsymbol{x}_M)\right) + 2\text{Cov}\left(\gamma^M V(\boldsymbol{x}_M), \sum_{m=0}^{M-1} \gamma^m \Delta V(\boldsymbol{z}_m)\right)$$
$$+ \text{Cov}\left(\sum_{m=0}^{M-1} \gamma^m \Delta V(\boldsymbol{z}_m), \sum_{m=0}^{M-1} \gamma^m \Delta V(\boldsymbol{z}_m)\right). \tag{35}$$

For the first term, linearity of the covariance in each argument together with the definition of $\sigma_V^2(\boldsymbol{x}_M)$ yields

$$\text{Cov}\left(\gamma^M V(\boldsymbol{x}_M), \gamma^M V(\boldsymbol{x}_M)\right) = \gamma^{2M} \sigma_V^2(\boldsymbol{x}_M). \tag{36}$$

For the second term, we obtain

$$\text{Cov}\left(\gamma^M V(\boldsymbol{x}_M), \sum_{m=0}^{M-1} \gamma^m \Delta V(\boldsymbol{z}_m)\right) = \sum_{m=0}^{M-1} \gamma^{M+m} \text{Cov}\left(V(\boldsymbol{x}_M), \Delta V(\boldsymbol{z}_m)\right)$$
$$\leq \sum_{m=0}^{M-1} \gamma^{M+m} \sqrt{\mathbb{V}[V(\boldsymbol{x}_M)]} \sqrt{\mathbb{V}[\Delta V(\boldsymbol{z}_m)]}$$
$$\leq \frac{1}{2} \sum_{m=0}^{M-1} \gamma^{M+m} \left(\mathbb{V}[V(\boldsymbol{x}_M)] + \mathbb{V}[\Delta V(\boldsymbol{z}_m)]\right)$$
$$\leq \frac{1}{2} \sum_{m=0}^{M-1} \gamma^{M+m} \left(\sigma_V^2(\boldsymbol{x}_M) + \sigma_\Delta^2(\boldsymbol{z}_m)\right)$$
$$\leq \frac{M\gamma^M}{2} \sigma_V^2(\boldsymbol{x}_M) + \frac{1}{2} \sum_{m=0}^{M-1} \gamma^{M+m} \sigma_\Delta^2(\boldsymbol{z}_m). \tag{37}$$

The first line follows from the linearity of the covariance in each argument, the second line is due to the Cauchy-Schwarz inequality, Young's inequality results in the third line, the fourth line is a consequence of the definitions of $\sigma_\Delta^2(\boldsymbol{x}_m)$ and $\sigma_V^2(\boldsymbol{x}_M)$, and the last line uses $\gamma^{M+m} \leq \gamma^M$. Similarly, we obtain for the last term in (35) that

$$\text{Cov}\left(\sum_{m=0}^{M-1} \gamma^m \Delta V(\boldsymbol{z}_m), \sum_{m=0}^{M-1} \gamma^m \Delta V(\boldsymbol{z}_m)\right) = \sum_{m=0}^{M-1} \sum_{m'=0}^{M-1} \gamma^{m+m'} \text{Cov}\left(\Delta V(\boldsymbol{z}_m), \Delta V(\boldsymbol{z}_{m'})\right)$$
$$\leq \sum_{m=0}^{M-1} \sum_{m'=0}^{M-1} \gamma^{m+m'} \sqrt{\mathbb{V}[\Delta V(\boldsymbol{z}_m)]} \sqrt{\mathbb{V}[\Delta V(\boldsymbol{z}_{m'})]}$$
$$\leq \frac{1}{2} \sum_{m=0}^{M-1} \sum_{m'=0}^{M-1} \gamma^{m+m'} \left(\mathbb{V}[\Delta V(\boldsymbol{z}_m)] + \mathbb{V}[\Delta V(\boldsymbol{z}_{m'})]\right)$$

using the linearity of the covariance in both arguments in the first line, the Cauchy-Schwarz inequality in the second line, and Young's inequality in the last line. Observe that

$$\sum_{m=0}^{M-1} \sum_{m'=0}^{M-1} \gamma^{m+m'} \mathbb{V}[\Delta V(\boldsymbol{z}_m)] = \sum_{m=0}^{M-1} \gamma^m \mathbb{V}[\Delta V(\boldsymbol{z}_m)] \sum_{m'=0}^{M-1} \gamma^{m'}$$
$$\leq \frac{1}{1-\gamma} \sum_{m=0}^{M-1} \gamma^m \mathbb{V}[\Delta V(\boldsymbol{z}_m)]$$
$$\leq \frac{1}{1-\gamma} \sum_{m=0}^{M-1} \gamma^m \sigma_\Delta^2(\boldsymbol{z}_m)$$

where the second line is due to $\sum_{m'=0}^{M-1} \gamma^{m'} \leq \sum_{m'=0}^{\infty} \gamma^{m'} = \frac{1}{1-\gamma}$ and the third line follows from the definition of $\sigma_\Delta^2(\boldsymbol{x}_m)$. Therefore, we have

$$\mathrm{Cov}\left(\sum_{m=0}^{M-1} \gamma^m \Delta V(\boldsymbol{z}_m), \sum_{m=0}^{M-1} \gamma^m \Delta V(\boldsymbol{z}_m)\right) \leq \frac{1}{1-\gamma} \sum_{m=0}^{M-1} \gamma^m \sigma_\Delta^2(\boldsymbol{z}_m). \tag{38}$$

Substituting (36), (37), (38) into (35), we obtain

$$\sigma_V^2(\boldsymbol{x}_0) \leq \gamma^{2M} \sigma_V^2(\boldsymbol{x}_M) + M\gamma^M \sigma_V^2(\boldsymbol{x}_M) + \sum_{m=0}^{M-1} \gamma^{M+m} \sigma_\Delta^2(\boldsymbol{z}_m) + \frac{1}{1-\gamma} \sum_{m=0}^{M-1} \gamma^m \sigma_\Delta^2(\boldsymbol{z}_m).$$

Finally, we have $\gamma^{2M} \leq \gamma^M$, such that we can simplify this expression to

$$\sigma_V^2(\boldsymbol{x}_0) \leq (M+1)\gamma^M \sigma_V^2(\boldsymbol{x}_M) + \frac{\gamma^M - \gamma^{M+1} + 1}{(1-\gamma)} \sum_{m=0}^{M-1} \gamma^m \sigma_\Delta^2(\boldsymbol{z}_m),$$

which concludes the proof.

$\square$

**Lemma B.3.** *Consider a sequence of training input sets* $\mathbb{X}_n = \{(\boldsymbol{x}_i)_{i=1,\dots,n}\}$ *with* $\mathbb{X}_n \subset \mathbb{X}_{n+1}$. *Let* $\sigma_{g,n}^2$ *be defined by* (6) *with training input set* $\mathbb{X}_n$ *and prior* $\mathcal{GP}(m_g, k_g)$. *Moreover, define* $[\boldsymbol{K}_g(\boldsymbol{X}_N)]_{i,j} = k_g(\boldsymbol{x}_i, \boldsymbol{x}_j)$. *Then, it holds that*

$$\log\det(\boldsymbol{I} + \sigma^{-2}\boldsymbol{K}_g(\boldsymbol{X}_N)) = \sum_{n=1}^{N} \log(1 + \sigma^{-2}\sigma_{g,n-1}^2(\boldsymbol{x}_n)). \tag{39}$$

*Proof.* Let $y_n = g(\boldsymbol{x}_n) + \epsilon_n$ for an arbitrary function $g \sim \mathcal{N}(m_g, k_g)$ and $\epsilon_n \sim \mathcal{N}(0, \sigma^2)$ is i.i.d. noise. Then, we have $\boldsymbol{y}_N | \boldsymbol{g}_N \sim \mathcal{N}(0, \sigma^2\boldsymbol{I})$ and $\boldsymbol{y}_N \sim \mathcal{N}(\boldsymbol{m}_g, \boldsymbol{K}_{\boldsymbol{g}_N} + \sigma^2\boldsymbol{I})$, where $[\boldsymbol{y}_N]_i = y_i$, $[\boldsymbol{g}_N]_i = g(\boldsymbol{x}_i)$ and $[\boldsymbol{m}_g]_i = m_g(\boldsymbol{x}_i)$. Therefore, it follows that

$$\frac{1}{2}\log\det(\boldsymbol{I} + \sigma^{-2}\boldsymbol{K}_g(\boldsymbol{X}_N)) = H(\boldsymbol{y}_N) - H(\boldsymbol{y}_N|\boldsymbol{g}_N),$$

where $H$ is the differential entropy. Due to the chain rule for differential entropies, we have

$$H(\boldsymbol{y}_N) = \sum_{n=1}^{N-1} H(y_{n+1}|\boldsymbol{y}_n) + H(y_1).$$

Moreover, it holds that $y_n|\boldsymbol{y}_{n-1} \sim \mathcal{N}(\mu_{g,n-1}(\boldsymbol{x}_n), \sigma_{g,n-1}^2(\boldsymbol{x}_n))$ such that

$$\sum_{n=1}^{N-1} H(y_{n+1}|\boldsymbol{y}_n) + H(y_1) = \frac{1}{2}\sum_{n=1}^{N} \log(\sigma^2 + \sigma_{g,n-1}^2(\boldsymbol{x}_n)).$$

Combining these identities, we obtain (39), which concludes the proof. $\square$

**Lemma B.4.** *Let* $M_n$ *be a sequence such that* $\gamma^{M_n} \leq \kappa/n^2$ *for* $\kappa \in (0,1)$. *Then,*

$$\sum_{n=1}^{\infty} (M_n + 1)\gamma^{M_n} \leq c_1 \tag{40}$$

*with*

$$c_1 = \frac{\kappa\pi^2(\log(\kappa) + \log(\gamma) - e^{-1}) - 6\kappa}{6\log(\gamma)}. \tag{41}$$

*Proof.* Since $\gamma^{M_n} \leq \kappa/n^2$, we have

$$\sum_{n=1}^{\infty} \gamma^{M_n} \leq \sum_{n=1}^{\infty} \frac{\kappa}{n^2}$$

$$\leq \frac{\kappa \pi^2}{6}. \tag{42}$$

Moreover, $\gamma^{M_n} \leq \kappa/n^2$ implies $M_n \geq \log(\kappa/n^2)/\log(\gamma)$. Let $\alpha(n) = M_n - \log(\kappa/n^2)/\log(\gamma)$. Therefore, we obtain

$$\sum_{n=1}^{\infty} M_n \gamma^{M_n} = \sum_{n=1}^{\infty} (\frac{\log(\kappa/n^2)}{\log(\gamma)} + \alpha(n)) \frac{\kappa}{n^2} \gamma^{\alpha(n)},$$

$$\leq \kappa \sum_{n=1}^{\infty} \frac{\log(\kappa/n^2)}{\log(\gamma)n^2} + \frac{\alpha(n)\gamma^{\alpha(n)}}{n^2},$$

where the second line follows from the fact that $\gamma^{\alpha(n)} \leq 1$. The second term can be bounded using

$$\max_{q \in \mathbb{R}} q\gamma^q \leq -\frac{1}{e\log(\gamma)}$$

which yields

$$\sum_{n=1}^{\infty} M_n \gamma^{M_n} \leq \frac{\kappa}{\log(\gamma)} \sum_{n=1}^{\infty} \frac{\log(\kappa) - 2\log(n)}{n^2} - \frac{1}{en^2}.$$

Furthermore, it can be shown that $\sum_{n=1}^{\infty} \log(n)/n^2 \leq 1$, such that

$$\sum_{n=1}^{\infty} M_n \gamma^{M_n} \leq \frac{\kappa}{\log(\gamma)} \left( \frac{\pi^2(\log(\kappa) - e^{-1})}{6} - 2 \right)$$

$$\leq \frac{\kappa\pi^2(\log(\kappa) - e^{-1}) - 12\kappa}{6\log(\gamma)}. \tag{43}$$

Combining (42) and (43) concludes the proof. $\square$

**Theorem 3.2.** *Consider a dynamical system* (1) *satisfying Assumption* 2.1*, such that Assumption* 2.2 *holds for the value function $V$. Assume that parameters $\boldsymbol{\theta}_{\mathrm{UCB}}^n$ are chosen via* (16) *with GP mean* (11) *and variance* (12) *learned from a data set $\mathbb{D}^n$ which is obtained from roll-outs of length $\gamma^{M_n} \leq \kappa/n^2$ for $\kappa \in (0,1)$ in each episode $n = 1, \ldots, N$. Then, the regret* (3) *satisfies*

$$R_N \leq c\sqrt{NM_N \Gamma_{k_\Delta}(N) \left(\Gamma_{k_\Delta}(NM_N) + \log(N)\right)} \tag{44}$$

*with probability $1 - \delta$ and constant $c \in \mathbb{R}_{\geq 0}$ for $N > 1$.*

*Proof.* Since we define $\beta_n$ using $\delta_n = 6\delta/\pi^2 n^2$ in every episode, Lemma B.1 together with the union bound yields

$$\sum_{n=1}^{N} J(\boldsymbol{\theta}^*) - J(\boldsymbol{\theta}_{\mathrm{UCB}}^n) \leq \sum_{n=1}^{N} 2\beta_{n-1}\mathbb{E}_{\boldsymbol{s}_0}\left[\sigma_{V,n-1}\left([\boldsymbol{s}_0; \boldsymbol{\theta}_{\mathrm{UCB}}^n]\right)\right], \tag{45}$$

$$\leq \mathbb{E}_{\boldsymbol{s}_0}\left[\sum_{n=1}^{N} 2\beta_{n-1}\sigma_{V,n-1}\left([\boldsymbol{s}_0; \boldsymbol{\theta}_{\mathrm{UCB}}^n]\right)\right], \tag{46}$$

where the second line follows from the linearity of the expectation. We expand the variances using Lemma B.2, such that we obtain

$$\sum_{n=1}^{N} J(\boldsymbol{\theta}^*) - J(\boldsymbol{\theta}_{\mathrm{UCB}}^n) \leq \mathbb{E}_{\boldsymbol{s}_0}\left[\sum_{n=1}^{N} 2\beta_{n-1}\left((M_n + 1)\gamma^{M_n}\sigma_{V,n-1}^2(\boldsymbol{x}_{M_n}^n)\right.\right.$$

$$\left.\left. + \frac{\gamma^{M_n} - \gamma^{M_n+1} + 1}{1 - \gamma} \sum_{m=0}^{M_n-1} \gamma^m \sigma_{\Delta,n-1}^2(\boldsymbol{z}_m^n)\right)^{\frac{1}{2}}\right] \tag{47}$$

with probability $1 - \delta$ for $\boldsymbol{x}_0^n = [\boldsymbol{s}_0; \boldsymbol{\theta}_{\text{UCB}}^n]$. Due to the Cauchy-Schwarz inequality, we can change the order between summation and square root resulting in

$$
\sum_{n=1}^{N} J(\boldsymbol{\theta}^*) - J(\boldsymbol{\theta}_{\text{UCB}}^n) \leq \mathbb{E}_{\boldsymbol{s}_0}\left[ 2\left( N \sum_{n=1}^{N} \beta_{n-1}^2 (M_n + 1)\gamma^{M_n}\sigma_{V,n-1}^2(\boldsymbol{x}_{M_n}^n) \right.\right.
$$
$$
\left.\left. + \frac{\gamma^{M_n} - \gamma^{M_n+1} + 1}{1-\gamma} \sum_{m=0}^{M_n-1} \gamma^m \beta_{n-1}^2 \sigma_{\Delta,n-1}^2(\boldsymbol{z}_m^n) \right)^{\frac{1}{2}} \right] \tag{48}
$$

with probability $1 - \delta$. Here, the index $n - 1$ in $\sigma_{V,n-1}^2$ and $\sigma_{\Delta,n-1}^2$ indicates that these are the posterior variances given the data set $\mathbb{D}^{n-1}$. We continue to analyze the two terms on the right side of this inequality separately. For the first term, observe that continuity of the kernel $k_V$ guarantees continuity of $\sigma_V^2$, such that the compact set $\mathcal{S}$ implies the existence of a finite value $\max_{\boldsymbol{x}\in\mathcal{X}}\sigma_{V,0}^2(\boldsymbol{x}) \geq \sigma_{V,n}^2(\boldsymbol{x})$ for all $\boldsymbol{x} \in \mathcal{X} = \mathcal{S} \times \Theta$. Therefore, Lemma B.4 ensures

$$
\sum_{n=1}^{N} \beta_{n-1}^2 (M_n+1)\gamma^{M_n}\sigma_{V,n-1}^2(\boldsymbol{x}_{M_n}^n) \leq \beta_N^2 c_1 \max_{\boldsymbol{x}\in\mathcal{X}}\sigma_{V,0}^2(\boldsymbol{x}). \tag{49}
$$

For the second term in (48), we first consider the sum over the sub-sequence $\boldsymbol{z}_0^n$. Moreover, let $\sigma_{\Delta,n,0}^2$ denote the variance based on training inputs $\boldsymbol{z}_0^1, \ldots, \boldsymbol{z}_0^n$, i.e., the sub-sequence of initial states in roll-out trajectories $\tau^n$. Since the GP posterior variance cannot decrease when removing data points (Vivarelli, 1998; Lederer, 2023), we obtain

$$
\sum_{n=1}^{N} \sigma_{\Delta,n-1}^2(\boldsymbol{z}_0^n) \leq \sum_{i=1}^{N} \sigma_{\Delta,n-1,0}^2(\boldsymbol{z}_0^n). \tag{50}
$$

We can upper bound the variances through through logarithmic terms via

$$
\sigma_{\Delta,n-1,0}^2(\boldsymbol{z}_0^n) \leq \sigma^2 \frac{\sigma^{-2}\sigma_{\Delta,n-1,0}^2(\boldsymbol{z}_0^n)}{\log(1+\sigma^{-2}\sigma_{\Delta,n-1,0}^2(\boldsymbol{z}_n^0))} \log(1+\sigma^{-2}\sigma_{\Delta,n-1,0}^2(\boldsymbol{z}_n^0)) \tag{51}
$$
$$
\leq c_2 \log(1 + \sigma^{-2}\sigma_{\Delta,n-1,0}^2(\boldsymbol{z}_n^0)) \tag{52}
$$

where the second line follows from the monotonous growth of $\frac{q}{\log(1+q)}$, such that

$$
c_2 = \frac{\max_{\boldsymbol{z}\in\mathcal{Z}}\sigma_{\Delta,0}^2(\boldsymbol{z})}{\log(1 + \sigma^{-2}\max_{\boldsymbol{z}\in\mathcal{Z}}\sigma_{\Delta,0}^2(\boldsymbol{z}))}. \tag{53}
$$

This allows us to apply Lemma B.3 to obtain

$$
\sum_{n=1}^{N} \gamma^0 \beta_{n-1}^2 \sigma_{\Delta,n-1}^2(\boldsymbol{z}_n^0) \leq c_2\beta_N^2 \log\det(\boldsymbol{I} + \sigma^{-2}\boldsymbol{K}_\Delta(\boldsymbol{Z}_N^0)), \tag{54}
$$

where $\boldsymbol{Z}_N^0$ is the concatenation of the sequence $\boldsymbol{z}_0^1, \ldots, \boldsymbol{z}_0^n$. Since this approach can be used analogously for all other subsequences $\boldsymbol{z}_m^n$ with $n = 1, \ldots, N$, we have the bound

$$
\sum_{n=1}^{N}\sum_{m=0}^{M_n-1} \gamma^m \beta_{n-1}^2 \sigma_{\Delta,n-1}^2(\boldsymbol{z}_0^n) \leq \sum_{m=0}^{M_N-1} c_2\beta_N^2 \log\det(\boldsymbol{I}+\sigma^{-2}\boldsymbol{K}_\Delta(\boldsymbol{Z}_N^m)) \tag{55}
$$

since $\gamma \leq 1$. We can upper bound the right-hand side by taking the maximum over the data sets in $\mathcal{Z}$ with $\mathcal{Z} = \mathcal{X} \times \mathcal{S}$, which results in

$$
\sum_{m=0}^{M_N-1} c_2\beta_N^2 \log\det(\boldsymbol{I}+\sigma^{-2}\boldsymbol{K}_\Delta(\boldsymbol{Z}_N^m)) \leq \max_{\substack{\boldsymbol{z}_m^n\in\mathcal{Z}, \\ m=0,\ldots,M_n-1, \\ n=1,\ldots,N}} \sum_{m=0}^{M_N-1} c_2\beta_N^2 \log\det(\boldsymbol{I}+\sigma^{-2}\boldsymbol{K}_\Delta(\boldsymbol{Z}_N^m)) \tag{56}
$$
$$
\leq 2c_2\beta_N^2 M_N \Gamma_{k_\Delta}(N) \tag{57}
$$

where we employ the maximum information gain $\Gamma_{k_\Delta}$ for the kernel $k_\Delta$ as defined in (7) and exploit its monotonicity. Thus, we have

$$\sum_{n=1}^{N} \sum_{m=0}^{M_n-1} \gamma^m \beta_{n-1}^2 \sigma_{\Delta,n-1}^2(\boldsymbol{z}_m^n) \leq 2c_2 \beta_N^2 M_N \Gamma_{k_\Delta}(N). \tag{58}$$

Substituting (49) and (58) into (48) results in

$$\sum_{n=1}^{N} (J(\boldsymbol{\theta}^*) - J(\boldsymbol{\theta}_{\text{UCB}}^n)) \leq \mathbb{E}_{\boldsymbol{s}_0} \left[ 2\beta_N \sqrt{N} \sqrt{c_3 + c_4 M_N \Gamma_{k_\Delta}(N)} \right], \tag{59}$$

where

$$c_3 = c_1 \max_{\boldsymbol{x} \in \mathcal{X}} \sigma_{V,0}^2(\boldsymbol{x}), \tag{60}$$

$$c_4 = \frac{4c_2}{(1-\gamma)}, \tag{61}$$

since $\gamma^{M_n} - \gamma^{M_n+1} \leq 1$. We further simplify this bound by noting that $\Gamma_{k_\Delta}(N) \geq \Gamma_{k_\Delta}(0) \geq \log(1 + \sigma^{-2} \max_{\boldsymbol{z} \in \mathcal{Z}} \sigma_{\Delta,0}(\boldsymbol{z}))$ for $N \geq 1$ due to Lemma B.3, such that we obtain

$$\sum_{n=1}^{N} (J(\boldsymbol{\theta}^*) - J(\boldsymbol{\theta}_{\text{UCB}}^n)) \leq \mathbb{E}_{\boldsymbol{s}_0} \left[ c_5 \beta_N \sqrt{N} \sqrt{M_N \Gamma_{k_\Delta}(N)} \right], \tag{62}$$

where

$$c_5 = 2\sqrt{\frac{c_3}{\log(1 + \sigma^{-2} \max_{\boldsymbol{z} \in \mathcal{Z}} \sigma_{\Delta,0}(\boldsymbol{z}))} + c_4}. \tag{63}$$

Since $\beta_N$ defined in (31) also depends on training data, we similarly proceed to bound its maximum over possible data sets by

$$\max_{\boldsymbol{z}_n \in \mathcal{Z}, n=1,\ldots,N} \beta_N \leq 2\sqrt{d_s L_k} B\lambda \sqrt{\Gamma_{k_\Delta}(NM_N) - 2\log\left(\frac{6\delta}{\pi^2 N^2}\right)} + \sigma B, \tag{64}$$

where the maximum information gain is with respect to the full data set with $\sum_{n=1}^{N} M_n \leq NM_N$ samples. The Cauchy-Schwarz inequality together with $\Gamma_{k_\Delta}(\sum_{n=1}^{N} M_n) \geq \log(1 + \sigma^{-2} \max_{\boldsymbol{z} \in \mathcal{Z}} \sigma_{\Delta,0}(\boldsymbol{z}))$ for $N \geq 1$ results in

$$\max_{\boldsymbol{z}_n \in \mathcal{Z}, n=1,\ldots,N} \beta_N \leq 2\sqrt{d_s L_k} B\lambda \left( \left( 2 + \frac{2\sigma^2 B^2}{\log(1 + \sigma^{-2} \max_{\boldsymbol{z} \in \mathcal{Z}} \sigma_{\Delta,0}(\boldsymbol{z}))} \right) \Gamma_{k_\Delta}(NM_N) \right.$$
$$\left. + 4\log\left(\frac{\pi^2 N^2}{6\delta}\right) \right)^{\frac{1}{2}}. \tag{65}$$

Due to $4\log\left(\frac{\pi^2 N^2}{6\delta}\right) = 4\log\left(\frac{\pi^2}{6\delta}\right) + 8\log(N)$ and the implication of $N \geq 2$ by the assumption of $N > 1$ for the integer $N$, we have $4\log\left(\frac{\pi^2}{6\delta}\right) + 8\log(N) \leq \left(4\log\left(\frac{\pi^2}{6\delta}\right)/\log(2) + 8\right)\log(N)$. Thus, we can further simplify the bound to

$$\max_{\boldsymbol{z}_n \in \mathcal{Z}, n=1,\ldots,N} \beta_N \leq c_6 \sqrt{\Gamma_{k_\Delta}(NM_N) + \log(N)} \tag{66}$$

with

$$c_6 = 2\sqrt{d_s L_k} B\lambda \sqrt{\max\left\{ \left( 2 + \frac{2\sigma^2 B^2}{\log(1 + \sigma^{-2} \max_{\boldsymbol{z} \in \mathcal{Z}} \sigma_{\Delta,0}(\boldsymbol{z}))} \right), \frac{4\log\left(\frac{\pi^2}{6\delta}\right)}{\log(2)} + 8 \right\}}. \tag{67}$$

Substituting (66) into (62) the argument of the expectation becomes independent of $s_0$ and we obtain

$$\sum_{n=1}^{N}(J(\boldsymbol{\theta}^*) - J(\boldsymbol{\theta}_{\text{UCB}}^n)) \leq c_5 c_6 \sqrt{NM_N\Gamma_{k_\Delta}(N)(\Gamma_{k_\Delta}(NM_N) + \log(N))}. \tag{68}$$

**Corollary 3.3.** *Assume that the assumptions of Theorem 3.6 are satisfied and $M_N \in \mathcal{O}(\log(N))$ holds. Then, with probability $1 - \delta$, the regret (3) satisfies*

$$R_N \in \mathcal{O}^* \left( \sqrt{N\Gamma_{k_\Delta}^2(N\log(N))} \right) \tag{69}$$

*where $\mathcal{O}^*$ denotes asymptotic expressions up to dimension-independent logarithmic factors.*

*Proof.* Due to Theorem 3.6, we have

$$R_N \in \mathcal{O}^* \left( \sqrt{NM_N\Gamma_{k_\Delta}(N)(\Gamma_{k_\Delta}(NM_N) + \log(N))} \right). \tag{70}$$

Using monotonicity of the maximum information gain and using $M_N \in \mathcal{O}(\log(N))$, this expression can be simplified to

$$R_N \in \mathcal{O}^* \left( \sqrt{N\log(N)\Gamma_{k_\Delta}^2(N\log(N)) + N\log(N)^2\Gamma_{k_\Delta}(N\log(N))} \right). \tag{71}$$

Since $\mathcal{O}^*$ notation denotes asymptotic expressions only up to dimension-independent logarithmic factors, we equivalently have

$$\mathcal{O}^* \left( \sqrt{N\log(N)\Gamma_{k_\Delta}^2(N\log(N)) + N\log(N)^2\Gamma_{k_\Delta}(N\log(N))} \right) = \mathcal{O}^* \left( \sqrt{N\Gamma_{k_\Delta}^2(N\log(N)) + N\Gamma_{k_\Delta}(N\log(N))} \right)$$

$$= \mathcal{O}^* \left( \sqrt{N\Gamma_{k_\Delta}^2(N\log(N))} \right), \tag{72}$$

which concludes the proof. $\square$

$\square$

## C. Bounding the Information Gain

**Lemma C.1.** *Define $\boldsymbol{\Xi} = \text{blkdiag}(\boldsymbol{\Xi}_1, \ldots, \boldsymbol{\Xi}_N)$, where $\boldsymbol{\Xi}_n \in \mathbb{R}^{M_n \times (M_n + 1)}$ is given by*

$$\boldsymbol{\Xi}_n = \begin{bmatrix} 1 & -\gamma & 0 & \ldots & 0 & 0 \\ 0 & 1 & -\gamma & \ldots & 0 & 0 \\ \vdots & \vdots & \vdots & \ddots & \vdots & \vdots \\ 0 & 0 & 0 & \ldots & 1 & -\gamma \end{bmatrix}. \tag{73}$$

*Moreover, let $\tilde{\boldsymbol{X}}_N$ denote the matrix of all $\boldsymbol{x}_m^n = [\boldsymbol{s}_m^n; \boldsymbol{\theta}^n]$ $m = 1, \ldots, M_n$, $n = 1, \ldots, N$, arranged in the order they occur in the trajectories. Then, we have the identity*

$$\boldsymbol{K}_\Delta(\boldsymbol{Z}_N)) = \boldsymbol{\Xi} K_V(\tilde{\boldsymbol{X}}_N)\boldsymbol{\Xi}^T. \tag{74}$$

*Proof.* To proof this identity, observe that we can express the temporal difference kernel $k_\Delta$ as the vector-matrix-vector product

$$k_\Delta(z_i, z_j) = \begin{bmatrix} 1 & -\gamma \end{bmatrix} \begin{bmatrix} k_Q(x_i, x_j) & k_Q(x_i, [s_j', \pi_{\theta_j}(s_j'), \theta_j]) \\ k_Q([s_i', \pi_{\theta_i}(s_i'), \theta_i], x_j) & k_Q([s_i', \pi_{\theta_i}(s_i'), \theta_i], [s_j', \pi_{\theta_j}(s_j'), \theta_j]) \end{bmatrix} \begin{bmatrix} 1 \\ -\gamma \end{bmatrix}.$$

Considering the matrix $\tilde{\boldsymbol{X}}_1$ consisting of a single trajectory, this immediately implies that we have

$$K_\Delta(\boldsymbol{Z}_1, \boldsymbol{Z}_1) = \boldsymbol{\Xi} K(\tilde{\boldsymbol{X}}_1, \tilde{\boldsymbol{X}}_1)\boldsymbol{\Xi}^T. \tag{75}$$

We do not have a coupling via a $\gamma$-term between trajectories, such that we obtain a block-diagonal structure for $\boldsymbol{\Xi}$ when considering multiple trajectories resulting in (75). $\square$

**Lemma C.2.** *A block-diagonal matrix $\boldsymbol{\Xi}$ with blocks* (73) *satisfies*

$$\|\boldsymbol{\Xi}\|^2 \leq (1+\gamma)^2, \tag{76}$$

*where $\|\boldsymbol{\Xi}\|$ is the spectral norm of $\boldsymbol{\Xi}$.*

*Proof.* The spectral norm of a matrix corresponds to its largest singular value, which in turn is the root of the maximum eigenvalue of $\boldsymbol{\Xi}\boldsymbol{\Xi}^T$. Thus, we have

$$\|\boldsymbol{\Xi}\|^2 = \lambda_{\max}(\boldsymbol{\Xi}\boldsymbol{\Xi}^T). \tag{77}$$

Since $\boldsymbol{\Xi}$ is block-diagonal, the matrix $\boldsymbol{\Xi}\boldsymbol{\Xi}^T$ is also block-diagonal, i.e. $\boldsymbol{\Xi}\boldsymbol{\Xi}^T = \text{blkdiag}(\boldsymbol{\Xi}_1\boldsymbol{\Xi}_1^T, \ldots, \boldsymbol{\Xi}_N\boldsymbol{\Xi}_N^T)$. Due to the block-diagonal structure of $\boldsymbol{\Xi}$, its eigenvalues are given by the eigenvalues of its blocks such that

$$\|\boldsymbol{\Xi}\|^2 = \max_{n=1,\ldots,N} \lambda_{\max}(\boldsymbol{\Xi}_n\boldsymbol{\Xi}_n^T). \tag{78}$$

Moreover, for each of its blocks, we can compute the matrix $\boldsymbol{\Xi}_n\boldsymbol{\Xi}_n^T$ in closed-form yielding

$$\boldsymbol{\Xi}_n\boldsymbol{\Xi}_n^T = \begin{bmatrix} 1+\gamma^2 & -\gamma & 0 & \ldots & 0 \\ -\gamma & 1+\gamma^2 & -\gamma & \ldots & 0 \\ \vdots & \vdots & \vdots & \ddots & \vdots \\ 0 & 0 & 0 & \ldots & 1+\gamma^2 \end{bmatrix}. \tag{79}$$

Due to (Kulkarni et al., 1999, Theorem 2.2), the maximum eigenvalue of such a tri-diagonal Toeplitz matrix is bounded by

$$\lambda_{\max}(\boldsymbol{\Xi}_n\boldsymbol{\Xi}_n^T) = 1 + \gamma^2 - 2\sqrt{\gamma^2}\cos\left(\frac{M_n\pi}{M_n+1}\right), \tag{80}$$

$$\leq 1 + \gamma^2 + 2\gamma, \tag{81}$$

$$\leq (1+\gamma)^2 \tag{82}$$

which concludes the proof. $\qquad\square$

**Lemma C.3.** *Let $\tilde{\boldsymbol{X}}_N$ denote the matrix of all $\boldsymbol{x}_m^n = [\boldsymbol{s}_m^n; \boldsymbol{\theta}^n]$ $m = 1, \ldots, M_n$, $n = 1, \ldots, N$. Then, it holds that*

$$\log\det(\boldsymbol{I} + \sigma^{-2}\boldsymbol{K}_\Delta(\boldsymbol{Z}_N)) \leq \log\det\left(\boldsymbol{I} + \frac{(1+\gamma)^2}{\sigma^2}\boldsymbol{K}_V(\tilde{\boldsymbol{X}}_N)\right). \tag{83}$$

*Proof.* Due to Lemma C.1, we have the identity

$$\log\det(\boldsymbol{I} + \sigma^{-2}\boldsymbol{K}_\Delta(\boldsymbol{Z}_N)) = \log\det(\boldsymbol{I} + \sigma^{-2}\boldsymbol{\Xi}\boldsymbol{K}_V(\tilde{\boldsymbol{X}}_N)\boldsymbol{\Xi}^T). \tag{84}$$

The log-determinant of a matrix corresponds to sum of the logarithm of its eigenvalues. Moreover, the eigenvalues $\lambda_i(\boldsymbol{I} + \boldsymbol{A})$ for an arbitrary quadratic matrix $\boldsymbol{A}$ are given by $1 + \lambda_i(\boldsymbol{A})$. Therefore, we obtain the identity

$$\log\det(\boldsymbol{I} + \sigma^{-2}\boldsymbol{K}_\Delta(\boldsymbol{Z}_N)) = \sum_{i=1}^{NM_N} \log(1 + \sigma^{-2}\lambda_i(\boldsymbol{\Xi}\boldsymbol{K}_V(\tilde{\boldsymbol{X}}_N)\boldsymbol{\Xi}^T)). \tag{85}$$

The eigenvalues of the matrix $\boldsymbol{\Xi}\boldsymbol{K}_V(\tilde{\boldsymbol{X}}_N)\boldsymbol{\Xi}^T$ correspond to its singular values since it is positive definite. Similarly, the eigenvalues of $\boldsymbol{K}_V(\tilde{\boldsymbol{X}}_N)$ correspond to its eigenvalues as it is also positive definite. Hence, the min-max characterization of singular values guarantees

$$\lambda_i(\boldsymbol{\Xi}\boldsymbol{K}_V(\tilde{\boldsymbol{X}}_N)\boldsymbol{\Xi}^T) \leq \|\boldsymbol{\Xi}\|^2\lambda_i(\boldsymbol{K}_V(\tilde{\boldsymbol{X}}_N)). \tag{86}$$

Substituting this bound into (85), we obtain

$$\log\det(\boldsymbol{I} + \sigma^{-2}\boldsymbol{K}_\Delta(\boldsymbol{Z}_N)) = \sum_{i=1}^{NM_N} \log(1 + \sigma^{-2}\|\boldsymbol{\Xi}\|^2\lambda_i(\boldsymbol{K}_V(\tilde{\boldsymbol{X}}_N))). \tag{87}$$

---

**Algorithm 2** GP-based Q-learning with parameterized policy adapted from (Chowdhury & Oliveira, 2023)

---

1: Initialize $\mathbb{D}^1$ (e.g., $\mathbb{D}^1 \leftarrow \emptyset$);
2: **for** $n = 1, \dots, N$ **do**
3:     $\hat{\boldsymbol{v}} = \boldsymbol{0}$
4:     **for** $i = 1, \dots, K$ **do**
5:         Obtain $\mu_V, \sigma_V^2$ using (5), (6) with $\hat{\boldsymbol{v}}$ as target and $\boldsymbol{S}$ as input
6:         $\hat{Q}(\boldsymbol{s}, \boldsymbol{a}) = r(\boldsymbol{s}, \boldsymbol{a}) + \gamma \mu_V([\boldsymbol{s}, \boldsymbol{a}] + \gamma \beta \sigma_V([\boldsymbol{s}, \boldsymbol{a}])$
7:         $\boldsymbol{\theta} \leftarrow \arg\max_\theta \mathbb{E}_{\boldsymbol{s}_0}[\hat{Q}(\boldsymbol{s}_0, \boldsymbol{\pi}_\theta(\boldsymbol{s}_0))]$;
8:         $\hat{\boldsymbol{v}} \leftarrow \hat{Q}(\boldsymbol{S}^+, \boldsymbol{\pi}_\theta(\boldsymbol{S}^+))$
9:     $\boldsymbol{\theta}_n \leftarrow \boldsymbol{\theta}$
10:    Roll-out $\boldsymbol{\pi}(\cdot, \boldsymbol{\theta}_n)$ for $M_n$ steps
11:    Measure $\tau^n = \{(\boldsymbol{s}_0^n, r_0^n), \dots, (\boldsymbol{s}_{M_n}^n, r_{M_n}^n)\}$
12:    Augment data set $\mathbb{D}^{n+1} \leftarrow \mathbb{D}^n \cup \tau^n$

---

Finally, we employ Lemma C.2, which results in

$$\log\det(\boldsymbol{I} + \sigma^{-2}\boldsymbol{K}_\Delta(\boldsymbol{Z}_N)) \leq \sum_{i=1}^{NM_N} \log(1 + \frac{(1+\gamma)^2}{\sigma^2}\lambda_i(\boldsymbol{K}_V(\tilde{\boldsymbol{X}}_N))), \tag{88}$$

$$\leq \log\det\left(\boldsymbol{I} + \frac{(1+\gamma)^2}{\sigma^2}\boldsymbol{K}_V(\tilde{\boldsymbol{X}}_N)\right), \tag{89}$$

where the second line follows from the equivalence of the log-determinant of a matrix and the sum over the logarithm of its eigenvalues. $\qquad\square$

**Theorem 3.4.** *The maximum information gain for the temporal difference kernel $k_\Delta$ is bounded by the information gain of its base kernel, i.e.,*

$$\Gamma_{k_\Delta}(NM_N) \in \mathcal{O}(\Gamma_{k_V}(NM_N)) \tag{90}$$

*for a non-decreasing sequence $M_n$, $n = 1, \dots, N$.*

*Proof.* We consider the scaling factor $(1+\gamma)^2$ on the right side of (83) as part of the variance, i.e., assume a noise variance $\tilde{\sigma}^2 = \frac{\sigma^2}{1+\gamma^2}$. Then, the right side corresponds to $\Gamma_{k_V}(N(M_N+1))$. As the scaling factor is constant, it does not influence the asymptotic behavior for growing $N$, which yields (90). $\qquad\square$

# D. Details of Numerical Experiments

We compare our TD-GP-UCB algorithm to two baselines:

- **MC-GP-UCB:** To demonstrate the benefits of exploiting the temporal difference decomposition of value functions in the GP model, we compare to a variant of GP-UCB that directly uses the Monte-Carlo value function estimates as training targets. For this, we perform Gaussian process regression on $V \sim \mathcal{GP}(m_V, k_V)$ using the dataset $\tilde{\mathbb{D}} = \{(\tilde{\boldsymbol{x}}_n, \tilde{y}_n)_{n=1,\dots,N}\}$, where $\tilde{\boldsymbol{x}}_n = [\boldsymbol{s}_0^n, \boldsymbol{\theta}^n]$ and $\tilde{y}_n = \sum_{t=0}^{M_n} \gamma^t r(\boldsymbol{s}_t^n, \boldsymbol{\pi}(\boldsymbol{s}_t^n, \boldsymbol{\theta}^n))$. This allows us to directly use (5) and (6) with $\tilde{\mathbb{D}}$ to compute the posterior, while no other changes are made compared to our implementation of Algorithm 1.
- **kQ-learn:** As an example for a variant of kernelized Q-learning, we adapt the approach for optimistic value iteration in (Chowdhury & Oliveira, 2023) to our problem setting as depicted in Algorithm 2. In contrast to the original algorithm from (Chowdhury & Oliveira, 2023), we discounted values are used in the iteration in Algorithm 2. Moreover, the number of value iterations can be set to an arbitrary number $K$. To avoid excessive computation times, we limit the number of value iterations to $K = 10$ per episode. Finally, we optimize over policy parameters $\boldsymbol{\theta}$ during value iteration instead of individual actions using the expectation of the optimistic Q-function over the initial state distribution as objective.
- **DDPG:** To illustrate benefits over modern reinforcement learning techniques, we use Deep Deterministic Policy Gradients (Lillicrap et al., 2016) as an example for the ubiquitous class of actor-critic methods. We use the implementation from (Navale, 2022), which topped the OpenAI Pendulum leaderboard (pen). To ensure fair comparison, we constrain the actor to linear policies with the same parameter bounds as our method.

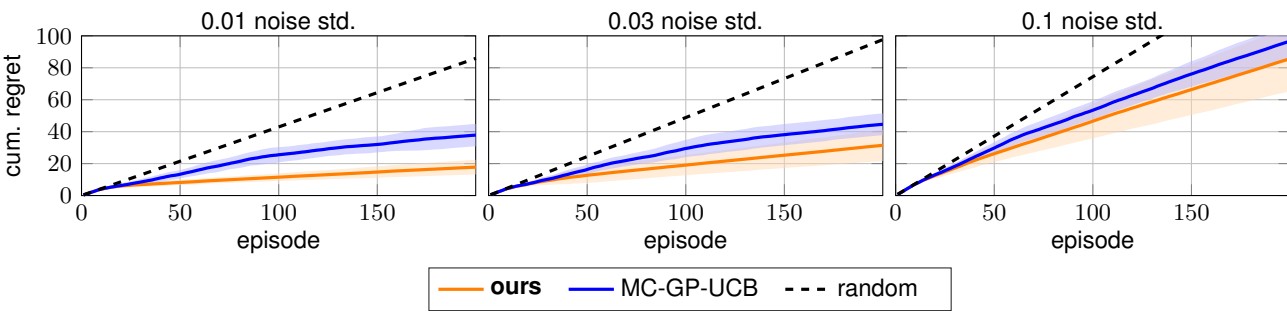

*Figure 6.* Average cumulative regret for the Gymnasium Pendulum with random initial state distribution, with Gaussian noise injected in the dynamics. Shaded areas illustrate one standard deviation confidence intervals.

## E. Discussion of Performance of Baselines

Despite empirically known strong performance (DDPG) and theoretical performance guarantees (kQ-learn), baselines seem to perform poorly in our numerical experiments. Therefore, we provide further reasoning to explain why their performance is not surprising.

**kQ-learn:** Finite horizon planning problems are known to require $K$ to be equal to the roll-out length (Chowdhury & Oliveira, 2023), i.e., $M_N = K$. More generally, RL theory suggests the necessity of large values $K$ to approximate the true infinite horizon discounted $Q$-function well. This indicates the necessity of $K > 200$ in our experiments. However, each iteration requires solving one generally non-convex optimization problem per data point. We empirically observe a linear growth of computation time with increasing $K$ in our experiments. This leads to an increase of the computation time of kQ-learn by a factor 10 for $K = 10$ compared to the proposed TD-GP-UCB, which clearly illustrates the intractability of significantly increasing $K$. When taking into account numerical optimizers for non-convex optimization problems, increasing the number of value iterations $K$ can also theoretically cause issues. Errors from sub-optimal optimization results get propagated through value iterations, such that they potentially accumulate significantly deteriorating the final policy. Finally, it should be noted that there is also no understanding how well the method can be expected to perform in practice to the best of our knowledge as this seems to be its first practical evaluation.

**DDPG:** We use an implementation of DDPG which is known to perform well in general (Navale, 2022), but we restrict its actor to linear policies, which can potentially affect the convergence rate. We keep the parameters at their default for the Pendulum environment as we also do not do any a priori parameter tuning for other methods, i.e., all hyperparameter tuning occurs within episodes that are depicted in the numerical evaluation. Finally, we want to emphasize that DDPG eventually converges to the optimal policy or a policy close to it, but it simply takes longer than the depicted 200 episodes. This 'zoom in' effect can give the impression of a worse performance than actually the case.

## F. Noisy Dynamics

We evaluated the methods on deterministic examples in Section 5 as the Pendulum and Cart Pole environments do not exhibit any process noise. In this section we report the performance of our method on a noisy version of the Pendulum environment with **random** initial states, in which i.i.d. Gaussian noise is injected in the dynamics (i.e. process noise $\boldsymbol{w} \sim \mathcal{N}(0, \sigma^2)$) across three different noise scales: $\sigma \in \{0.01, 0.03, 0.1\}$. The cumulative regret achieved in these environments is illustrated in Figure 6.

Our method TD-GP-UCB outperforms the baseline MC-GP-UCB across all noise scales, though the performance gap reduces through increasing noise scale. Note that the regret values across noise scales may not be comparable since $\boldsymbol{\theta}^*$ may differ across the scales. Nonetheless, when compared to the deterministic variant in Figure 2 (left) the regret values of TD-GP-UCB relative to that of the policy with random parameters seems to degrade increasingly with larger noise scales.

Further, both TD-GP-UCB and MC-GP-UCB exhibit linear growth of cumulative regret at the larger noise scale ($\sigma = 0.1$, Figure 6 (right)) for the depicted range of episodes. This can likely be attributed to two factors. On the one hand, data with higher noise scales is less informative and convergence will take longer in general, such that 200 episodes might not be enough to achieve convergence anymore. On the other hand, there is a larger distribution shift at test time compared to the deterministic variant, as we use a fixed set of randomly sampled initial states for evaluation that differ from those of the rollouts.

