# OpenReview forum: "Policy Search via Bayesian Optimization with Temporal Difference Gaussian Processes"
_ICML.cc/2026/Conference — ICML 2026 regular_

### Official Review · Reviewer_t6CU · 2026-02-19

**Soundness:** 3
**Presentation:** 3
**Significance:** 3
**Originality:** 3
**Overall Recommendation:** 4
**Confidence:** 1

**Summary:**

The authors propose a new Bayesian optimization approach using temporal difference learning with Gaussian process regression. They derive learning error bounds for this method, enabling cumulative regret analysis through upper confidence bounds, further refined by bounding maximal information gain. Various benchmarks taken for the study demonstrate its practical effectiveness.

**Compliance With Llm Reviewing Policy:**

Affirmed.

**Final Justification:**

I had very few concerns about the submission, and the authors kindly addressed them. I can see that the submission was positively assessed by 2 other reviewers. I trust their evaluations and don't mind the submission being accepted.

**Key Questions For Authors:**

As I mentioned above, the manuscript falls out of my expertise, and here I can only point at something insignificant, such as:

1) Line 034 - should you re-introdice abbreviation of GP as it was never mentioned before (the abstract does not count)?

2) Fig. 1 can be improved in terms of clarity (e.g. colorbars).

**Limitations:**

Yes

**Strengths And Weaknesses:**

It appears to be difficult for me to assess this submission as the topic is mismatch with my field of research. Therefore I do not stand for the scores I put below.

Although, it is still clear to me that the authors thoroughly make a comparison with the baselines, also challenging ones, and are open about the limitations of their approach.

---

> ### Author Rebuttal · Authors · 2026-03-30
>
> Thank you for your assessment, despite the paper falling outside of your field of expertise. We have directly applied your feedback to the manuscript, and updated the introduction as well as Figure 1 to make it more accessible.

---

> > ### Author Rebuttal · Reviewer_t6CU · 2026-04-01
> >
> > I had very few concerns about the submission, and the authors kindly addressed them. I can see that the submission was positively assessed by 2 other reviewers. I trust their evaluations and don't mind the submission being accepted.

---

### Official Review · Reviewer_2aCm · 2026-03-12

**Soundness:** 3
**Presentation:** 3
**Significance:** 3
**Originality:** 3
**Overall Recommendation:** 5
**Confidence:** 3

**Summary:**

This paper proposes a policy search approach that combines Bayesian optimization with temporal difference Gaussian processes. The main motivation of the paper is that standard Bayesian optimization does not use important structural information in the policy since it treats the policy evaluation as a black-box scalar objective. However, rollouts generate structured sequential data which can be used through temporal differences relationships to build a structured Gaussian process model of the value function. The paper demonstrates that using such structural information helps to learn return much more efficiently from the trajectory data. The paper provides regret analysis demonstrating that the regret grows sublinearly and the proposed approach converges to the optimal policy. The approach is tested on cart pole and pendulum environments and shows that the TD structure improves data efficiency (it’s faster).

**Compliance With Llm Reviewing Policy:**

Affirmed.

**Key Questions For Authors:**

Comments: Since each rollout contributes $M_n$ transitions to the GP model the dataset size grows as $\sum_n M_n$.  How does the computational cost would scale in practice for more episodes?

The method assumes access to an oracle that globally optimizes the acquisition function. It would be nice if the authors can elaborate more on this and how this can be met in practice.

Can the proposed approach scale to higher dimensional and nonlinear policy parameterizations such as NN policies?

**Limitations:**

yes

**Strengths And Weaknesses:**

Strength: The paper is very well-written and organized, I really enjoyed reading it. The results are clear and the main strength of the paper is the combination of TD structure with Bayesian optimization with strong theoretical guarantees. The results of the paper are really relevant to the literature of policy optimization, Bayesian optimization and RL.

Weakness: My main concern is the scalability of the approach that involves GP regression over trajectory transitions which is limited to low-dimensional policy parameterization and may not be practical for high dimensional setting and deep-RL. This is a limitation of GPs in general so it does not only affect the result of the paper.

---

> ### Author Rebuttal · Authors · 2026-03-30
>
> We are grateful for the positive assessment and the thoughtful questions.
>
> **Computational cost**
> If we use exact GP regression, we suffer from the standard cubic complexity, which yields $\mathcal{O}((nM_n)^3)$ in each episode $n$ due to the fast growing dataset size as correctly stated by the reviewer. For small experiments, this is still tractable in principle: $M_n\le 200$ and $N=200$ episodes results in $\le 40,000$ samples -- with sufficient RAM and a powerful CPU, exact inverses of this size can be computed within ~1h. However, in practice it is more convenient to employ one of the many computationally efficient GP approximations, see (Liu et al., 2020) for an overview. Since the proposed TD-GP remains a GP, many of these existing approximations are readily applicable. Due to its simplicity and very low complexity, we employ a spectral feature approximation (Rahimi & Recht, 2008) in our numerical experiments, see Section 5.1. With our naive implementation, this approximation already reduces the complexity to $\mathcal{O}(nM_n)$ in each episode $n$, but we could further reduce to $\mathcal{O}(M_n)$ using iterative updates. Thus, computational complexity of GP regression is not a critical issue in our implementation. To clarify this point, we will expand our discussion of this topic in Remark 3.2.
>
> **Oracle for global optimization**
> The requirement of an oracle that outputs an exact global maximum is merely a technical requirement. Without it, even perfect knowledge of the value function does not yield 0 regret. Therefore, this assumption is present in almost all literature on BO, even though it is not necessarily stated explicitly. It can be seen in the proof of Lemma B.1 that we can easily relax the requirement to an oracle that ouputs parameters $\theta_{UCB}$ with $J(\theta^*)-J(\theta_{UCB})<\epsilon$ similarly as in (Kirschner et al., 2019). If the parameter $\epsilon$ is a design choice of the oracle, we can let it shrink at the same rate as the remaining regret bound, leaving our result unaffected. This is a property that numerical optimization approaches can verifiably achieve, e.g., the simple approach of gridding, whose scalability can be improved via adaptive techniques that maintain the guarantees (Jones et al., 1993). It is straightford to see that simple strategies such initializing local optimizers with random starts (Danilova et al., 2022) can ensure these guarantees probabilistically. However, note that an increase in accuracy inevitably comes at the cost of run-time. Thus, we use a fixed number of 10 random starts with the Nelder-Mead algorithm as optimizer, which we empirically found to be sufficient for our experiments. A brief version of this discussion will be added to the updated version.
>
> **Scalability**
> As discussed in the rebuttal for reviewer gKD4, we believe the scalability issues to be tied to the global optimization problem, which renders the application of Algorithm 1 to high-dimensional policy parameterizations a challenge. We believe our contribution to be valuable due the many applications with low-dimensional parameterizations as mentioned by reviewer gKD4, see e.g. (Calandra et al, 2015; Fiducioso et al., 2019; Paulson et al., 2022; Berkenkamp et al., 2023). Moreover, we are convinced that modifications can enable an application to higher dimensional problems, e.g., certifiable NN policy fine-tuning. We can significantly reduce the sample complexity of GP regression via informative task-specific priors, which can be obtained by learning an input transform (Rai et al., 2019) or the entire kernel (Rothfuss et al., 2023) using a NN. Additionally, we can employ techniques to compactly represent NN policies via so called fingerprints (Harb et al., 2020) to reduce the dimensionality of the search space in optimization. Finally, we can adapt our approach to focus on local optimality along the lines of (Müller et al., 2021; Nguyen et al., 2022), such that local model accuracy suffices. While we believe that these changes will significantly improve scalability, they would require an extension of our theoretical analysis, which goes significantly beyond our current results. Hence, we leave this setting for future work. A sketch of these ideas will be added to the conclusion in the updated version.
>
> Kirschner et al. (2019). Adaptive and safe Bayesian optimization in high dimensions via one-dimensional subspaces. International Conference on Machine Learning, 3429-3438
>
> Danilova et al. (2022). Recent theoretical advances in non-convex optimization. High-Dimensional Optimization and Probability: With a View Towards Data Science, 79–163, Springer
>
> Jones et al. (1993). Lipschitzian optimization without the Lipschitz constant. Journal of Optimization Theory and Applications, 79(1), 157-181
>
> Rothfuss et al. (2023). Scalable PAC-Bayesian meta-learning via the PAC-optimal hyper-posterior: From theory to practice. Journal of Machine Learning Research, 24(386), 1-62

---

> > ### Author Rebuttal · Reviewer_2aCm · 2026-04-03
> >
> > I thank the authors for addressing all my concerns.

---

### Official Review · Reviewer_gKD4 · 2026-03-13

**Soundness:** 4
**Presentation:** 4
**Significance:** 4
**Originality:** 3
**Overall Recommendation:** 5
**Confidence:** 2

**Summary:**

The paper proposes a Bayesian optimization method for policy search that utilizes the sequential structure of on-policy trajectories instead of treating each evaluation as a black box in an infinite-horizon discounted-reward setting where value functions are modelled with a temporal-difference Gaussian process. The paper derives regret guarantees for a UCB-type Bayesian optimization algorithm, and provides an analysis of the corresponding information gain. In addition to the theoretical guarantees, the performance of the algorithm is evaluated empirically in comparison with standard Bayesian optimization baselines.

**Compliance With Llm Reviewing Policy:**

Affirmed.

**Key Questions For Authors:**

- In Theorem 3.3, a sequence for the trajectory length $M_n$ is provided and the regret result is proved under this sequence. How sensitive is the performance of the proposed method to this choice?
- The paper acknowledges that the proposed algorithm is efficient for low-dimensional policy parameterizations. For high-dimensional problems, what is the main bottleneck and where is the computational efficiency lost? A brief discussion on this would be useful for the readers.
- The value function $\hat{J}\_n(\theta)$ in (15) is averaged over the initial state distribution $s\_0\sim\rho$. In frequentist policy gradient methods, the initial state distribution has a fundamental impact on the optimality gap in terms of the concentrability coefficient $\|\|d_\mu\^{\pi^\star}/\rho\|\|_\infty$. Could it be possible to theoretically quantify the impact of $\rho$ on the regret for this method?

**Limitations:**

See the "Strengths and Weaknesses" section.

**Strengths And Weaknesses:**

**Strengths:**
- The key strength of the paper is the GP formulation, which enables a GP prior on the value function and TD-GP construction. This is a very clean way to inject the sequential structure into the BO framework, going beyond the black-box approaches. The idea of utilizing the the temporal/sequential structure of the on-policy trajectories is very significant as it yields data reuse for statistical efficiency within the Bayesian optimization framework across episodes.
- The paper is very well-written, and the positioning in the literature is very clear.
- The proposed algorithm is supported by theoretical as well as empirical guarantees, and achieves good performance.

**Weakness:**
- One potential weakness could be the dimensionality, as also acknowledged by the authors. On the other hand, given the applications of low-dimensional parameterizations, the contributions are still quite strong.

---

> ### Author Rebuttal · Authors · 2026-03-30
>
> We thank the reviewer for the positive feedback and the thoughtful questions.
>
> **Sensitivity to roll-out length**
> Letting the roll-out length grow faster than $\mathcal{O}(\log n)$ increases the growth rate of our regret bound. However, this is merely an artifact of our analysis as discussed at the end of Section 3.5, which can be similarly found in related model-based RL approaches. At its core, this artifact stems from the delay between GP updates and data collection. Longer roll-outs cause a larger delay, and thus increase the impact of this artifact. However, we do not observe a growing regret with growing roll-out length in practice. Choosing smaller roll-out lengths will have negative impacts when they become too small since they directly affect the residual error and its asymptotics (see Lemma B.4 and (49)). Therefore, the performance of our method is sensitive to the roll-out length in principle, but this dependency is not critical. The reason for this is that we can easily determine what constitutes too small because the residual is bounded by $\gamma^M k_V(x,x)$. For example, we employ $M_n=\lceil\frac{10}{1-\gamma} + \log n\rceil$ in our experiments, such that $\gamma^{M_n} \le e^{-10} \le 10^{-4}$ renders the residual practically irrelevant. We will add a short explanation on the effect of the roll-out length in the updated version.
>
> **Limited scalability to higher dimensions**
> When scaling to higher dimensional problems, most challenges relate to the inherent hardness of globally optimal decision making. Even if we had a closed-form expression for the value function, (approximately) finding the global maximum of a non-convex high dimensional function is a difficult task. Numerical solvers suffer from the curse of dimensionality, such that the sub-optimality of their outputs increases or their run-time grows (see the rebuttal to reviewer 2aCm for a discussion of global optimization). Naturally, this adversely affects the scalability of our approach to high-dimensional policy parameterizations, as it requires global optimization in every episode. Additionally, global optimization explores the full parameter space, which requires learning a global model of the value function. Without structural knowledge about the value function, this learning problem is naturally hard in high dimensions. In particular, we have an exponentially increasing data requirement with growing dimensionality. Note that the computational complexity of GP regression is not an issue by itself due to the availability of computationally efficient approximations (see rebuttal to reviewer 2aCm). Thus, we believe that the limited scalability is fundamentally tied to the problem of finding globally optimal parameters, rather than our specific approach. This belief is supported by the observation that scalable algorithms usually focus on simplified problems, e.g., local optimality with deep RL algorithms such as DDPG (Lilicrap et al., 2016). For BO approaches, scalable algorithms for various such simplified settings have been developed, e.g., focusing on local optimality (Müller et al., 2021), assuming additional structure on the objective (Mutny & Krause, 2019), or imposing additional cost on queries (Xie et al., 2024). We believe that these problem formulations can be a promising direction towards a scalable extension of our approach (see the rebuttal to reviewer 2aCm for more details). A brief version of this discussion will be added to the updated version.
>
> **Impact of initial state distributions**
> In our analysis, we do not see a dependency on the initial state distribution as the argument of the expectation over initial states is bounded independent from the states (see (68)). Regarding the specific relation to the concentrability coefficient $\|d_\mu^{\pi*}/\rho\|_\infty$, we are not exactly sure what the reviewer refers to. Coefficients of this form arise with PG methods, but we found $\mu$ to denote the initial distribution in this context (Agarwal et al., 2019, Section 4.1.1). $\rho$ is the data distribution, not the initial state distribution as in our work. Since $\rho$ changes in each episode, this approach only seems to make sense in an offline setting. However, analyzing the impact of the initial state distribution in the offline setting is out of the scope of this work.
>
> Mutny & Krause (2019). Efficient high dimensional Bayesian optimization with additivity and quadrature Fourier features. Advances in Neural Information Processing Systems, 9005-9016
>
> Xie etl al. (2024). Cost-aware Bayesian optimization via the pandora's box Gittins index. Advances in Neural Information Processing Systems, 115523-115562
>
> Agarwal et al. (2019). Reinforcement learning: Theory and algorithms. CS Dept, UW Seattle, WA, USA, Tech Rep, 32, 96

---

> > ### Author Rebuttal · Reviewer_gKD4 · 2026-04-03
> >
> > Thank you for your answers. My questions were addressed. I keep my score as is.

---

### Official Review · Reviewer_P4dz · 2026-03-15

**Soundness:** 3
**Presentation:** 3
**Significance:** 2
**Originality:** 2
**Overall Recommendation:** 4
**Confidence:** 3

**Summary:**

The paper proposes leveraging rollouts of dynamical systems to learn value functions using a temporal-difference Gaussian process and embedding this model into a Bayesian optimization policy search routine that explicitly exploits the temporal structure of the problem. The Bayesian posterior over the value function is then used to estimate an upper confidence bound on the return in a UCB-style algorithm. The authors also provide new analysis of the regret and the maximum information gain based on the modified GP structure. The evaluation is done on pendulum, cart-pole and LQR problems.

**Compliance With Llm Reviewing Policy:**

Affirmed.

**Final Justification:**

The literature cited by the authors makes a convincing case for the relevance of the targeted application domains, although I find the “90%” figure in their rebuttal substantially overstated. Given that and the fact that paper appears to be sound, I am happy to reconsider my score.

I hope the authors will follow through on their commitment to strengthen the numerical evaluation in the revised version. I would also encourage the authors to open-source their code, as this would make it easier for others to engage with and build on their contribution.

**Key Questions For Authors:**

See comments on significance

**Limitations:**

yes

**Strengths And Weaknesses:**

**Presentation**

The paper is well-written, notation is clear, contributions are stated at the beginning of the paper, and related concepts are introduced and explained early in the text.

**Soundness**

The work appears to be technically sound, the problem statement and the assumptions are stated clearly. However, I did not scrutinize the proofs in detail.

**Novelty and Significance**

My impression is that embedding the temporal-difference condition into the GP is done in a clever way, but beyond that I find that the overall approach appears largely incremental. While the theoretical analysis is valuable, the empirical evidence is not particularly convincing.

My understanding is that a Bayesian optimization approach should at least be more sample efficient, but this advantage is not clearly demonstrated in the evaluation, which I find contrived. In addition, the smoothness assumptions, which generally shouldn't hold neither for the pendulum or cart-pole value functions, together with the limited scalability in higher dimensions cast doubt on the significance of the proposed approach.

---

> ### Author Rebuttal · Authors · 2026-03-30
>
> We are thankful for the comments and want to clarify the mentioned issues.
>
> **Incremental contribution**
> We respectfully disagree with the reviewer's assessment of an incremental contribution as integrating the temporal difference structure into GPs is only a small part of our work. The majority of our contribution lies in handling the ramifications of this integration: we use values of $\Delta V$ as training target, but predict values for $V$. This disalignment between targets and predictions renders existing regret bounds and proof techniques for BO inapplicable to our approach. We overcome this limitation by developing new proof techniques that exploit the temporal difference structure to relate the GP variance for $V$ with the GP variance for $\Delta V$ along roll-outs (Lemma B.2). Only with such substantial innovations we are able to prove Theorem 3.3, which may appear similar to existing results at the surface. Similarly, existing bounds for the maximum information gain are limited to common kernels and do not apply to our custom kernel, but only with such a result, regret bounds like (18) become informative. Our analysis addresses this issue by introducing a different representation of the temporal difference kernel allowing us to exploit properties of Toeplitz matrices. This novel approach is crucial for making existing information gain bounds readily applicable to our problem. We believe that the novel ideas underlying the proofs of Theorems 3.3 and 3.5 are not just relevant in our particular setting, but can find broad application when dealing with composed kernels. Finally, we want to highlight that our numerical evaluation is also a relevant contribution as related work often entirely lacks experiments (Jin et al., 2020; Yang et al., 2020; Zhou et al., 2021; Chen et al., 2022; Chowdhury & Oliveira, 2023).
>
> **Unconvincing empirical evidence**
> We cannot follow the reviewer's critique. Importantly, our figures depict regret, i.e., slowly increasing curves indicate a high sample efficiency. Thus, our simulation results clearly demonstrate the superior sample efficiency of our approach throughout experiments. It achieves the lowest regret, i.e., the highest average rewards, accross all baselines for all experiments except for the Cart-Pole system, for which only MC-GP-UCB exhibits lower but increasing regret (i.e., not converged yet). More widely, BO methods (MC-GP-UCB, TD-GP-UCB) consistently outperform other baselines in terms of average sample efficiency. The non-BO baseline DDPG barely manages to achieve an improvement over random exploration in most of the benchmarks within the considered small number of episodes. Finally, we want to clarify that VI-GP-UCB is a kernelized Q-learning method. While related to BO, it is not considered a BO approach in literature, and consequently does not contradict the reviewer's intuition. To avoid confusion, we will refer to this method as kQ-learning in the updated version.
>
> **Impractical smoothness assumption**
> We appreciate the concern regarding the smoothness requirements, but we believe that it might arise from a misunderstanding. We can confirm that *optimal value functions*
> $$V^\star(s)=\max_a r(s,a)+\gamma\mathbb{E}_{s^+|a}[V^\star(s^+)]$$
> are usually not differentiable. However, we do not consider optimal value functions, but parameterized state-value functions
>
> $$V(s,\theta)=r(s,\pi(s,\theta))+\gamma\mathbb{E}_{s^+|\pi(s,\theta)}[V(s^+,\theta)],$$
>
> where $\theta$ is the parameter of the policy $\pi(s,\theta)$. Differentiability with respect to the policy parameter $\theta$ clearly holds under the assumptions of the policy gradient theorem, with extensions to higher order derivative following similarly. To understand smoothness with respect to states $s$, we employ the common fixed point definition of $V$, for which we can easily observe that
> $$\frac{\partial}{\partial s} V_{i+1}(s,\theta)= \frac{\partial}{\partial s} r(s,\pi(s,\theta)) + \gamma\int V_i(s^+,\theta)\frac{\partial}{\partial s}p(s^+|s,\pi(s,\theta)) d s^+,$$
> and similar identities for higher order derivatives. Thus, the smoothness of each $V_i$ is determined by the smoothness of the reward, the density and the policy. This property extends to value functions $V$ under the common convergence conditions on the sequence $V_i$. Hence, sufficient smoothness of the value function is not overly restrictive in general. We want to stress that smoothness is also implicitly required in much of the related work when using smooth kernels or features, see e.g. (Lazaric et al., 2012; Tu & Recht, 2018; Yang et al., 2020; Yeh et al., 2023;  Chowdhury & Oliveira, 2023). Moreover, our experiments clearly demonstrate that this assumption is not crucial in practice, which we attribute to the robustness of BO type algorithms under model misspecification (Bogunovic & Krause, 2021) as discussed in Section 5.3.
>
> **Limited scalability to higher dimensions**
> We refer to the rebuttals for reviewers gKD4 and 2aCm.

---

> > ### Author Rebuttal · Reviewer_P4dz · 2026-04-01
> >
> > I appreciate the authors’ additional clarification of their contribution, and I acknowledge the novelty and significance of the theoretical analysis. I also admit that my initial review was too short and not specific enough. My use of the term _incremental_ may have been misplaced.
> >
> > My concerns are mainly about the paper’s impact, particularly in terms of the problem settings targeted in the evaluation section. While I agree that the empirical results are consistent with the claims made in the paper, I am not fully convinced of their significance when they are demonstrated only on relatively simple benchmarks, that are further modified to better suit the proposed method.
> >
> > The paper targets robotics and control applications, a domain in which modern deep RL methods already perform extremely well, including in high-dimensional settings. In that context, the decision to evaluate mainly on Pendulum and CartPole environments is difficult to justify. I also find the comparison against DDPG unconvincing. DDPG is no longer generally regarded as a state-of-the-art baseline compared with methods such as TD3 or SAC. Moreover, constraining DDPG to use linear policies does not seem to be a fair comparison. The main strength of these methods is precisely their ability to handle large neural function approximators and scale to higher-dimensional problems, which is exactly where Bayesian optimization and Gaussian processes appear to struggle in practice.
> >
> > So despite the positive evaluation in this controlled and limited low-dimensional setting, I do not think it is reasonable to extrapolate and expect the advantage to persist on more general problems. There are also some minor concerns with the evaluation. The reported results are averaged over only five random seeds. Additionally, the evaluation does not appear to test the method in settings with stochastic transitions, which can affect the quality of temporal-difference targets.
> >
> > With all due respect and acknowledgement for the authors’ hard work, I am also skeptical of the discussion around scalability. In my view, it has become common for Gaussian process papers to defer scalability to future work, yet such promises are often not realized in practice. As a result, I would find claims of broader practical relevance more convincing if they were supported by stronger empirical evidence.
> >
> > All that said, I am not opposed to this work being accepted, but I stand by my skepticism regarding its impact and significance.

---

> > > ### Author Response · Authors · 2026-04-03
> > >
> > > We sincerely thank the reviewer for the clarification of concerns and pointing out weaknesses in our evaluation.
> > >
> > > **Significance and impact**
> > > We agree with the reviewer that deep RL methods generally perform well in high-dimensional settings. However, we want to emphasize that this is not the focus of our work, which addresses policy search problems for low-dimensional, structured policy parameterizations as emphasized throughout the paper. These structured parameterizations such as LQR and PID remain the backbone of modern control systems despite the developments in deep RL. For example, it is commonly estimated that more than 90% of industrial control systems employ PID (Hägglund & Guzman, 2024). This prevalence of low-dimensional policy parameterizations naturally causes a need for flexible and efficient parameter tuning methods, which is increasingly addressed using BO techniques in domains such as robotics (Jaquier et al., 2022; Widmer et al., 2023; Le et al., 2025), manufacturing systems (Kavas et al., 2025; Nobar et al., 2025; Li et al., 2026) and process control (Paulson et al., 2022; Krishnamoorthy, 2025; Richter et al., 2025). Importantly, these developments are application-driven and originate from the respective scientific communities illustrating the relevance of methods that are effective at handling low-dimensional policy parameterizations. Our approach is designed with such application-driven problems in mind, for which we believe our low-dimensional benchmarks with linear policies to be a reasonable choice. Thus, we are convinced that our results demonstrate the significance of our approach.
> > >
> > > **Discussion about scalability**
> > > We understand the reviewer's concern about deferring scalability, but it does not diminish the significance and relevance of our approach in the light of the previous paragraph. Scaling our approach to high dimensions essentially requires a change in the problem setting as discussed in the rebuttal to reviewer gKD4. Hence, we leave it for future work. We suggest specific modifications proposed in literature, which are known to address key scalability limitations of our current approach. We did not claim that such a modification would make our approach competitive to or even better than deep RL in high-dimensional settings. More specifically, we did not even suggest it as an alternative to deep RL in general, but highlighted fine-tuning as a potential application. Considering fine-tuning problems that require hardware experiments, we encounter scenarios in which deep RL is known to struggle in general, while the strengths of BO techniques can and have been leveraged (He et al., 2025). Hence, we believe that our discussion of the future potential for scalability is well-supported by related work and nuanced, without making promises that are essential for the paper's contribution.
> > >
> > > **Evaluation weaknesses**
> > > As discussed before, linear policies are a reasonable choice as parameterization motivated by their widespread practical usage. Moreover, it is pointless if a NN policy performs better if the application does not admit it, e.g., because of common restrictions to linear policies by practitioners. Thus, we believe that it is crucial to restrict all methods to this policy parameterization. Additionally, note that regret can only be compared within the same policy parameterization. We will increase the number of seeds and add ablation studies to demonstrate the impact of noise in the updated version. Note that we have done some preliminary evaluations with noise early on that indicated no relevant change in the trends of curves.
> > >
> > >
> > > Hägglund & Guzman (2024). Give us PID controllers and we can control the world. IFAC-PapersOnLine, 58(7), 103-108
> > >
> > > Jaquier et al. (2022). Geometry-aware Bayesian optimization in robotics using Riemannian Matérn kernels. Conference on Robot Learning, 794-805
> > >
> > > Widmer et al.(2023). Tuning legged locomotion controllers via safe bayesian optimization. Conference on Robot Learning, 2444-2464
> > >
> > > Le et al. (2025). Controller adaptation via learning solutions of contextual Bayesian optimization. IEEE Robotics and Automation Letters
> > >
> > > Kavas et al. (2025). In-situ controller autotuning by Bayesian optimization for closed-loop feedback control of laser powder bed fusion process. Additive Manufacturing, 99, 104641
> > >
> > > Nobar et al. (2024). Guided Bayesian optimization: Data-efficient controller tuning with digital twin. IEEE Transactions on Automation Science and Engineering, 22, 11304-11317
> > >
> > > Li et al. (2026). Bayesian optimization with active constraint learning for advanced manufacturing process design. IISE Transactions, 58(3), 257-271
> > >
> > > Krishnamoorthy, D. (2025). ECCBO: An inherently safe Bayesian optimization with embedded constraint control for real-time process optimization. Journal of Process Control, 152, 103467
> > >
> > > Richter et al. (2025). Bayesian optimization for automatic tuning of a MIMO controller of a flotation bank. Journal of Process Control, 147, 103388

---

### Decision · Program_Chairs · 2026-04-30

**Decision:**

Accept (regular)

**Comment:**

The paper describes a method using rollouts to learn value functions with a temporal-difference GP, and uses this model in a BO policy search.  The reviewers agreed that the paper was well written and appears sound.  The main objection had to do with significance, largely because the method seems most appropriate to relatively low-dimensional settings (in comparison with deep RL).  The authors replied that the low-dimensional setting remains important in a number of applications, and gave several examples.  There was a robust discussion, and at the end of the discussion the reviewers all favored acceptance, including reviewer P4dz (who initially was the most skeptical).